# Noradrenergic arousal after encoding reverses the course of systems consolidation in humans

Valentina Krenz[1], Tobias Sommer[2], Arjen Alink [2], Benno Roozendaal[3,4] & Lars Schwabe [1]✉

It is commonly assumed that episodic memories undergo a time-dependent systems consolidation process, during which hippocampus-dependent memories eventually become reliant on neocortical areas. Here we show that systems consolidation dynamics can be experimentally manipulated and even reversed. We combined a single pharmacological elevation of post-encoding noradrenergic activity through the $\alpha_2$-adrenoceptor antagonist yohimbine with fMRI scanning both during encoding and recognition testing either 1 or 28 days later. We show that yohimbine administration, in contrast to placebo, leads to a time-dependent increase in hippocampal activity and multivariate encoding-retrieval pattern similarity, an indicator of episodic reinstatement, between 1 and 28 days. This is accompanied by a time-dependent decrease in neocortical activity. Behaviorally, these neural changes are linked to a reduced memory decline over time after yohimbine intake. These findings indicate that noradrenergic activity shortly after encoding may alter and even reverse systems consolidation in humans, thus maintaining vividness of memories over time.

[1] Department of Cognitive Psychology, Institute of Psychology, Universität Hamburg, Von-Melle-Park 5, 20146 Hamburg, Germany. [2] University Medical Centre Hamburg-Eppendorf, Department of Systems Neuroscience, Martinistraße 52, 20246 Hamburg, Germany. [3] Department of Cognitive Neuroscience, Radboud University Medical Center, 6500 HB Nijmegen, The Netherlands. [4] Donders Institute for Brain, Cognition and Behaviour, Radboud University, Kapittelweg 29, 6525 EN Nijmegen, The Netherlands. ✉email: lars.schwabe@uni-hamburg.de

With time, episodic memories may undergo a neural reorganization. Specifically, the temporally graded amnesia in patients such as H.M. and neuroimaging findings suggested that memories are initially critically dependent on the hippocampus but, over time, relocated to neocortical areas during a process of systems consolidation[1–6]. This time-dependent neural reorganization of the memory trace may be accompanied by a semantization[7,8], and hence areas implicated in semantic memory, such as the ventromedial prefrontal cortex (vmPFC) and the inferior frontal gyrus (IFG)[5,9,10], are prime candidates for neocortical storage sites. Although this semantization over time may be adaptive in that it promotes the building of abstract knowledge structures[10,11], keeping specific and vivid memories may be particularly important for emotionally arousing events. However, whether the dynamics of systems consolidation may be shaped by environmental conditions, such as emotional arousal, remains unknown.

Stress and emotional arousal are powerful modulators of memory[12–16]. Extensive evidence demonstrates that arousal-induced noradrenergic activation of the basolateral amygdala (BLA) modulates neuroplasticity processes in other brain regions[17–20]. Most studies investigating noradrenergic arousal effects on memory have focused on episodic or contextual memories that depend on the hippocampus[21]. Surprisingly, however, the long-term fate of such memories and potential changes in systems consolidation processes remained completely unclear. A recent study in rodents provided first evidence that noradrenergic arousal shortly after encoding may prolong hippocampal involvement in long-term memory and hence alter systems consolidation[22]. This study showed that the administration of norepinephrine into the BLA shortly after training on an inhibitory avoidance discrimination task resulted in significantly increased episodic-like memory after a delay of 28d, compared to a saline administration. Even more strikingly, this study indicated that norepinephrine after encoding did not only maintain hippocampal dependency of memory after 28d but even led to an increased hippocampal dependency of memory over time, suggesting not only that systems consolidation processes can be experimentally manipulated, but that noradrenergic activation during initial consolidation might even reverse systems consolidation dynamics. Whether noradrenergic arousal can influence the dynamics of systems consolidation of memories in humans remains completely unknown.

In the present experiment, we aimed to unravel the impact of noradrenergic stimulation on systems consolidation and long-term memory maintenance in humans. To this end, participants encoded a series of pictures in an MRI scanner. Participants received orally either a placebo (PLAC) or the $\alpha_2$-adrenoceptor antagonist yohimbine (YOH) shortly before encoding. Immediate free recall was tested to ensure that initial memory encoding was comparable between groups. Critically, in order to probe time-dependent systems consolidation, delayed memory performance was tested either 1d or 28d after encoding, again in an MRI scanner, which enabled us to directly assess changes in the neural architecture of memory from encoding to test using univariate as well as multivariate functional MRI (fMRI) analyses. We predicted that YOH administration would enhance memory performance after 28d and decelerate or even reverse systems consolidation, as reflected by an increased hippocampal but reduced neocortical, in particular vmPFC and IFG, involvement. Moreover, leveraging multivariate pattern analysis for the assessment of encoding-retrieval similarity, we hypothesized that the representational pattern of memories at the remote test should become even more similar to the pattern at encoding, when noradrenergic stimulation was elevated after encoding.

As predicted, we show here that increased noradrenergic arousal shortly after encoding critically altered the systems consolidation dynamics. Whereas participants in the PLAC group show the expected systems consolidation process, with decreased hippocampal and increased neocortical activity over time, this process is reversed in the YOH group. Participants treated with YOH show increased hippocampal and reduced neocortical activity from 1d to 28d after encoding. Moreover, hippocampal encoding-retrieval similarity decreases from the 1d to the 28d test in the PLAC group but even increases in the YOH group. These neural changes are accompanied by a reduced decline of memory over time in participants that had received YOH. Together, these findings show that noradrenergic arousal shortly after encoding may not only alter but even reverse the dynamics of systems consolidation over time.

## Results

**Effective manipulation of arousal after encoding**. To determine the effect of post-encoding noradrenergic arousal on time-dependent systems consolidation in humans, we used a two-day fully crossed between-subjects design with the factors drug (PLAC vs. YOH) and delay (1d vs. 28d), resulting in four experimental groups: 1d/PLAC, 28d/PLAC, 1d/YOH, and 28d/YOH. On the first experimental day, participants ($n = 104$) received orally either a PLAC or 20 mg of the $\alpha_2$-adrenoceptor antagonist YOH right before they entered the MRI for encoding. The dosage and timing of drug administration was chosen based on the known pharmacodynamics of YOH[23,24], in order to achieve increased noradrenergic arousal shortly after the encoding session, i.e. during initial consolidation. To track the action of the drug, blood pressure was measured before and at six different time points (35, 55, 70, 85, 100, and 115 min) after drug administration. The efficacy of drug manipulation was tested using mixed-model ANOVAs with the between-subjects factors drug and delay and the within-subject factor time. Groups had comparable blood pressure before the drug administration (all $t < -0.61$, all $p > 0.242$, all $d < 0.12$). We found a significant drug × time interaction for both systolic ($F_{5.12,501.39} = 14.86$, $p < 0.001$, $\eta_p^2 = 0.13$) and diastolic ($F_{5.25,514.73} = 9.36$, $p < 0.001$, $\eta_p^2 = 0.09$) blood pressure (Fig. 1B). Importantly, even directly after encoding (and before the immediate free recall test) there was no effect of YOH on blood pressure (all $t < -1.06$, all $p > 0.111$, all $d < 0.32$), indicating that YOH was not yet effective during encoding. YOH did, however, increase both systolic and diastolic blood pressure from 85 min after drug administration until the end of day 1 (systolic: all $t > 2.89$, all $p < 0.005$, all $d > 0.57$; diastolic: all $t > 2.93$, all $p < 0.005$, all $d > 0.58$), thus showing the action of the drug shortly after encoding.

**Successful memory encoding**. Within 5 min after drug administration, participants encoded 60 pictures (30 neutral, 30 emotionally negative) in the MRI scanner, each presented once in each of three consecutive runs. To control for alertness during encoding, participants were instructed to respond to the fixation cross shown between trials with a button press. On average, participants missed only 1.44 (SD = 3.20) responses across all trials and runs, without any differences between groups (all $\beta < 0.54$, $p > 0.187$), suggesting that participants of all four groups remained attentive throughout the encoding task.

To further control for potential group differences in initial encoding, we asked participants to recall as many of the pictures as possible immediately after the encoding session. In this immediate free recall test, participants recalled on average 31.38 (SD = 9.58) of the 60 previously presented items. Although the

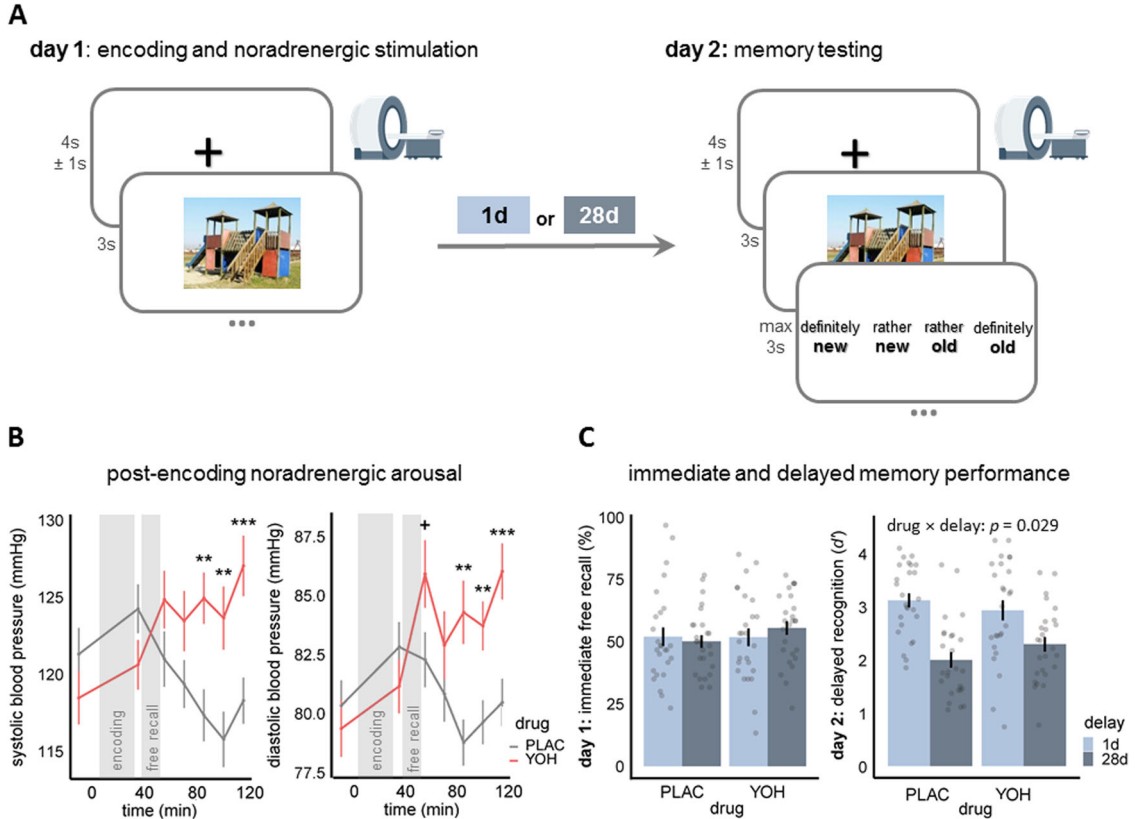

**Fig. 1 Experimental design, physiological, and behavioral results. A** Participants were tested on two experimental days: day 1, stimulus encoding and pharmacological manipulation of post-encoding noradrenergic activity and day 2, memory recognition. Both encoding and test took place in the MRI scanner. Critically, to investigate time-dependent consolidation processes, the memory test took place either 1d or 28d after encoding. The image of the playground is licensed under Creative Commons License; courtesy of Tomasz Sienicki (https://commons.wikimedia.org/wiki/File:Playground_29_ubt.JPG; image unchanged). **B** Effective manipulation of noradrenergic arousal after encoding: While groups did not differ at baseline (all $p > 0.242$, two-tailed Welch's $t$-tests) or shortly after encoding (all $p > 0.111$, two-tailed Welch's $t$-tests), participants of the yohimbine (YOH) group had significantly higher systolic (all $p < 0.005$, two-tailed Welch's $t$-tests) and diastolic (all $p < 0.005$, two-tailed Welch's $t$-tests) blood pressure from 85 min after drug intake until the end of experimental day 1 (drug × time: all $p < 0.001$, mixed ANOVAs). **C** A generalized linear mixed model (LMM) with the between-factors drug and delay and the within-factor emotion revealed no group-difference in immediate free recall performance on day 1, suggesting that encoding was comparable in the four groups. However, while memory performance significantly decreased from 1d to 28d after encoding (main effect delay: $\beta = -1.12$, $p < 0.001$, LMM), post-encoding noradrenergic arousal reduced this time-dependent memory decline (drug × delay: $\beta = 0.64$, $p = 0.029$, LMM): The YOH group showed a significantly smaller decrease in memory performance from 1d to 28d than the placebo (PLAC) group. All $n = 104$ participants. Bars represent mean ± SEM. Source data are provided as Source data file. **$p < 0.010$; ***$p < 0.001$.

delay to the recognition test should not be relevant for performance immediately after encoding, we ran a trial-wise binomial generalized linear mixed model (LMM) with drug (PLAC vs. YOH), delay (1d vs. 28d) and emotion (neutral vs. negative) and their interactions as fixed effects and the random intercept of participants and stimuli to not only assess potential drug effects on encoding but to also rule out potential differences between the 1d- and 28d-groups in initial encoding. This analysis showed a significant effect of the factor emotion ($\beta = 0.68$, $p < 0.001$, $z = 3.1$; supplementary Fig. 1), indicating overall higher free recall performance for emotionally negative compared to neutral stimuli. Critically, there was no effect of drug ($p = 0.499$), delay ($p = 0.403$) or drug × delay ($p = 0.281$), showing that the drug administration did not influence initial memory encoding and that the four groups had comparable memory performance shortly after stimulus encoding (Fig. 1C).

**Noradrenergic stimulation reduces time-dependent memory decline.** To examine the impact of noradrenergic stimulation on time-dependent changes in memory, we tested participants'

memory in a recognition test that took place either 1d or 28d after encoding, again in the MRI scanner. Before this recognition test, groups had comparable blood pressure, confirming that the drug was not active at the time of memory testing (all $F_{1,100} < 1.02$, all $p > 0.386$, $\eta_p^2 < 0.02$).

Overall, participants correctly recognized 85.11% (SD = 13.21%) of the old items (hits) and incorrectly classified only 4.21% (SD = 6.36%) of the new pictures as old (false alarms), demonstrating a high memory performance in the recognition test. Participants' intact memory for the learned items was further confirmed by the sensitivity index $d'$, which takes the individual response bias into account[25] and likewise indicated that memory performance was overall high (mean $d' = 2.59$, SD = 0.89). To test for time-dependent effects of noradrenergic stimulation on memory performance, $d'$-values were analyzed by means of an LMM with drug (PLAC vs. YOH), delay (1d vs. 28d), emotion (neutral vs. negative) and their interactions as fixed effects and the random intercept of participants. This analysis showed, as expected, that memory performance was lower at 28d than 1d after encoding ($\beta = -1.12$, 95%-$CI[-1.53, -0.72]$, $t_{125.82} = -5.43$, $p < 0.001$). This time-dependent decrease in $d'$ was smaller for

negative compared to neutral pictures (emotion × delay: $\beta = 0.42$, 95%-CI[0.14, 0.70], $t_{100} = 2.98$, $p = 0.003$; supplementary Fig. 2). Most importantly, there was a significant drug × delay interaction ($\beta = 0.64$, 95%-CI[0.06, 1.21], $t_{125.82} = 2.18$, $p = 0.029$), showing that the memory decline from 1d to 28d was weaker in the YOH group than in the PLAC group (Fig. 1C), irrespective of the emotionality of the encoded stimuli (drug × delay × emotion: $\beta = -0.31$, $p = 0.124$).

Participants' responses in the recognition test included ratings of confidence (Fig. 1A). An additional trial-wise generalized LMM on confidence for hits with drug (PLAC vs. YOH), delay (1d vs. 28d), emotion (neutral vs. negative) and their interactions as fixed effects and the random intercept of participants and stimuli revealed, as expected, a decrease in confidence in the 28d group, compared to the 1d group ($\beta = -1.91$, $p < 0.001$, $z = -5.98$). This decrease in confidence in recognizing old items was significantly lower for emotionally negative stimuli (emotion × delay: $\beta = 0.70$, $p = 0.007$, $z = 2.69$), but not influenced by noradrenergic stimulation (drug × delay: $\beta = 0.63$, $p = 0.151$, $z = 1.44$). No other main or interaction effects reached significance in this analysis (all $p > 0.110$). Moreover, in an additional analysis we weighted participants' responses by the level of confidence. This analysis indicated, as before, a significant decrease in memory performance in the 28d-group relative to the 1d-group (main effect delay: $\beta = -1.25$, 95%-CI[−1.68,−0.82], $t_{123.33} = -5.71$, $p < 0.001$). This time-dependent decrease in memory was again significantly lower in the YOH group than in the PLAC group (drug × delay: $\beta = 0.62$, 95%-CI[0.01,1.22], $t_{123.33} = 1.98$, $p = 0.0495$). Note that in none of these analyses the interaction drug × delay × emotion approached statistical significance (all $p > 0.094$).

Finally, although our study did not focus on potential differences between men and women and was not sufficiently powered to detect such effects, in light of findings suggesting sex differences in the impact of arousal or stress mediators on memory[26], we exploratively analyzed potential sex differences. Including the factor sex into the above LMM did not reveal a significant main effect of sex ($\beta = 0.07$, $p = 0.820$) nor any interactions with any other factors (all $p > 0.384$), suggesting that the effect of post-encoding noradrenergic arousal on memory performance over time was comparable in men and women.

**Noradrenergic stimulation increases hippocampal but decreases neocortical contributions to remote memory.** To determine the influence of post-encoding noradrenergic activation on systems consolidation, we measured BOLD-activity during both encoding and recognition testing after 1d and 28d, respectively. Our neural analyses focused mainly on the hippocampus, which had been at the center of the research on systems consolidation[1–6]. Our univariate fMRI analysis revealed a significant drug × delay interaction for hippocampal activity for old vs. new pictures (SVC peak-level: x = 22, y = −38, z = 4, t = 3.31, $p$(FWE) = 0.036, k = 10). As shown in Fig. 2A, while hippocampal activity for old (vs. new) pictures tended to be reduced 28d relative to 1d after encoding in the PLAC group ($t_{49.70} = 1.75$, $p = 0.086$, $d = 0.49$), hippocampal activity significantly increased 28d vs. 1d after encoding in participants who had received YOH ($t_{45.05} = -3.54$, $p < 0.001$, $d = 0.98$). Moreover, at the 28d test, hippocampal activity was significantly higher in the YOH than in the PLAC group ($t_{48.24} = 2.53$, $p = 0.015$, $d = 0.70$).

While we predicted a time-dependent decrease in hippocampal activation in the PLAC group, we analyzed also activity in the IFG and vmPFC, two neocortical regions that are known to be of particular relevance for remote, semantic memory[5,9,10] and in which thus activity might increase over time. In sharp contrast to

the pattern observed in the hippocampus, the IFG showed a significant increase for old (vs. new) pictures from 1d to 28d in the PLAC group ($t_{43.16} = -3.61$, $p < 0.001$, $d = 1.00$), whereas there was no such increase in IFG activity in participants who had received YOH ($t_{46.37} = 0.50$, $p = 0.620$, $d = 0.14$; drug × delay, SVC peak-level: x = −44, y = 32, z = 12, t = 4.01, $p_{corr}$(FWE) = 0.042, k = 62; Fig. 2B). Interestingly, activation of the IFG was negatively correlated with memory performance expressed as sensitivity index $d'$ across groups ($t_{102} = -2.22$, $r = -0.21$, $p = 0.029$; Fig. 2C), suggesting that the decline of memory performance over time was directly associated with the increased IFG involvement in memory. This correlation remained significant after removing outliers, which were defined in accordance to Tukey's method[27]. There were no effects of drug × delay in the vmPFC or in an exploratory whole-brain analysis.

While the previous analysis focused on brain activity for old vs. new items during memory testing, in a next step we analyzed changes in brain activity from the last run of encoding to recognition testing either 1d or 28d later. By taking explicitly the activity at encoding into account, this analysis provides insights into dynamic changes in memory-related activity over time and its modulation by noradrenergic arousal. We focused specifically on changes relative to activity in the last run of the encoding task since this run reflected not only encoding activity but due to the preceding stimulus presentations also immediate memory-related activity. We found a significant drug × delay interaction for recognition vs. encoding for the IFG (SVC peak-level: x = −48, y = 34, z = 12, t = 5.6, $p_{corr}$(FWE) < 0.001, k = 842; whole-brain peak-level: x = −48, y = 34 z = 12, t = 5.6, $p$(FWE) = 0.002, k = 45). As displayed in Fig. 2D, while the PLAC group showed a significant increase in IFG activity from encoding to retrieval at 1d vs. 28d ($t_{47.02} = -4.71$, $p < 0.001$, $d = 1.31$), in the YOH group there was even a decrease in IFG activity from encoding to retrieval with increasing retention delay ($t_{44.86} = 2.69$, $p = 0.010$, $d = 0.75$). Interestingly, an exploratory whole-brain analysis also revealed a significant drug × delay interaction for recognition vs. encoding in the same direction for the precuneus (drug × delay, whole-brain peak-level: x = −4, y = −50, z = 46, t = 5.21, $p$(FWE) = 0.009, k = 27), showing a significant increase in activity after a short compared to a long retention delay in the PLAC group ($t_{48.69} = -3.56$, $p < 0.001$, $d = 0.99$), in line with recent findings that identified the precuneus as a site for neocortical long-term storage[28,29], but a significant decrease in activity for the YOH group ($t_{47.12} = 3.95$, $p < 0.001$, $d = 1.10$). No drug × delay interactions were observed for the vmPFC or the hippocampus in this analysis.

An additional analysis of potential group differences during the final encoding run did not indicate a main effect of drug, neither in any of our a-priori defined regions of interest (ROIs) nor in an exploratory whole-brain analysis, thus providing further evidence that the drug administration left the encoding activity itself unaffected and that YOH was active only after encoding, in line with our autonomic measures.

**Noradrenergic stimulation reverses the time-dependent changes in IFG-hippocampus connectivity.** In a next step, we performed a psychophysiological interaction (PPI) analysis to test whether the functional connectivity of the hippocampus with the IFG during the recognition task changed as a function of time and noradrenergic stimulation. Using the IFG as seed, this analysis showed that hippocampal-IFG connectivity was significantly increased at 28d relative to 1d after encoding in the PLAC group ($t_{42.94} = -3.08$, $p = 0.004$, $d = 0.85$), whereas there was even a significant decrease ($t_{45.78} = 2.40$, $p = 0.020$, $d = 0.67$) in hippocampal-IFG functional connectivity at 28d vs. 1d in the

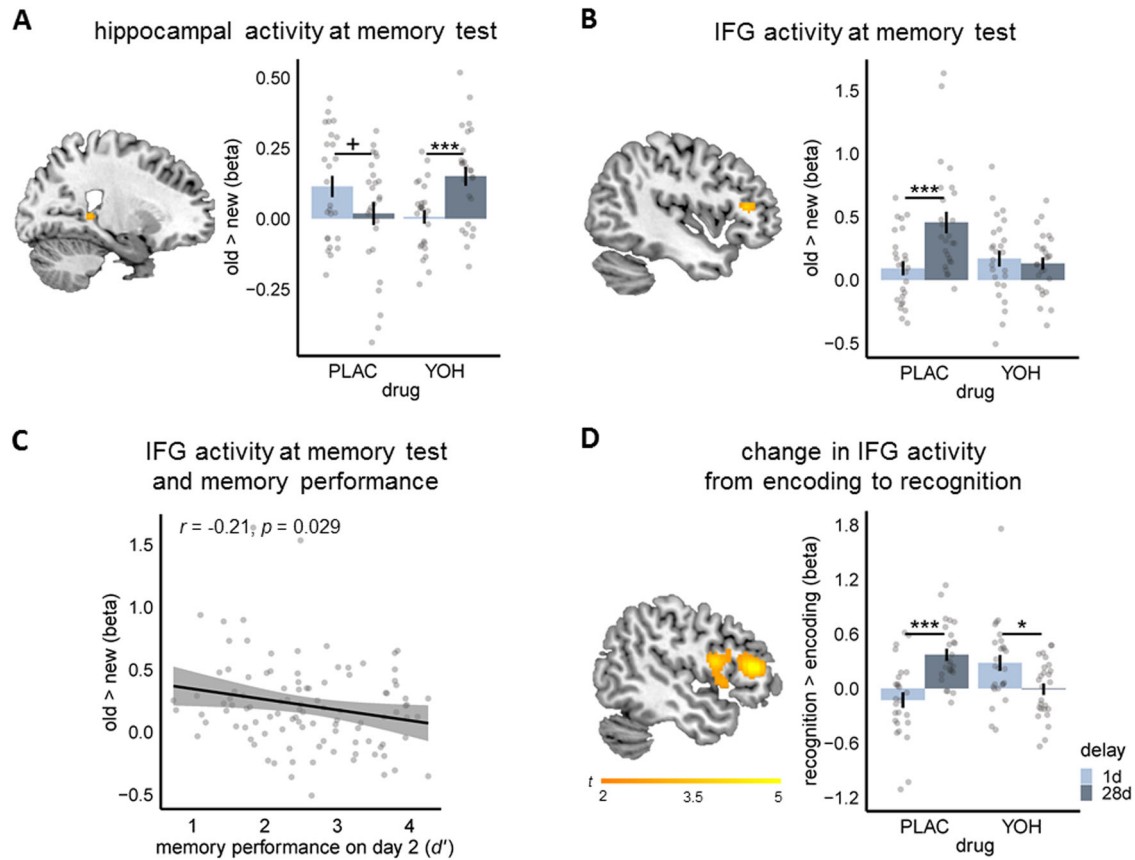

**Fig. 2 Noradrenergic stimulation increases hippocampal but decreases neocortical contributions to remote memory. A** While hippocampal activity tended to decrease from 1d relative to 28d in the placebo (PLAC) group ($p = 0.086$, two-tailed Welch's $t$-test), there was even a significant increase in hippocampal activity during memory testing from 1d to 28d in the yohimbine (YOH) group ($p < 0.001$, two-tailed Welch's $t$-test; drug × delay, SVC peak-level: $x = 22$, $y = −38$, $z = 4$, $p(\text{FWE}) = 0.036$, mixed ANOVA). **B** Conversely, inferior frontal gyrus (IFG) activity increased significantly from 28d relative to 1d in the PLAC group ($p < 0.001$, two-tailed Welch's $t$-test) but not in the YOH group ($p = 0.620$, two-tailed Welch's $t$-test; SVC peak-level: $x = −44$, $y = 32$, $z = 12$, $p_{corr}(\text{FWE}) = 0.042$, mixed ANOVA). **C** Pearson correlation analysis indicated that IFG activity at 28d-delayed memory test was negatively associated with memory performance on day 2. Note that this correlation remained significant after removing outliers from the analysis. **D** Moreover, while there was a significant increase in IFG activity from encoding to memory testing at the 28d vs. 1d-delayed test in the PLAC group ($p < 0.001$, two-tailed Welch's $t$-test), there was even a significant decrease in IFG activity from encoding to retrieval with increasing retention delay in the YOH group ($p = 0.010$, two-tailed Welch's $t$-test; drug × delay, SVC peak-level: $x = −48$, $y = 34$, $z = 12$, $p_{corr}(\text{FWE}) < 0.001$, mixed ANOVA). Bonferroni correction was applied for the number of regions of interest in each analysis. All $n = 104$ participants. Visualizations show $t$-maps for the interesting contrasts superimposed on sagittal sections of T1-weighted template images and beta-values for the significant cluster. Bars represent mean ± SEM. Source data are provided as Source data file. $^{+}p < 0.100$; $^{*}p < 0.050$; $^{***}p < 0.001$.

YOH group (drug × delay, SVC peak-level: $x = −22$, $y = −40$, $z = −2$, $t = 3.77$, $p(\text{FWE}) = 0.009$, $k = 10$; Fig. 3). As an increase in hippocampal-IFG connectivity has been linked with the generation of semantic associations[30], this finding further indicates that noradrenergic activation after encoding may reverse systems consolidation processes.

**Noradrenergic stimulation increases pattern reinstatement in the hippocampus over time.** Successful remembering has been associated with the reinstatement of brain activity present during encoding at test[2,31–33]. To determine the influence of norepinephrine and time on the reactivation of encoding-related activation patterns during memory testing, we assessed in a final step Encoding-Retrieval-Similarity (ERS) as a multivariate measure of trial-specific episodic reinstatement[34–39] applying a searchlight-based representational similarity analysis (RSA) approach[40–42]. Because a decrease in memory reinstatement is thought to reflect a more abstract memory representation and that the episodic details of a specific memory are not successfully

retrieved[43], we expected a decrease in similarity between activation patterns during encoding and memory testing, i.e. ERS, during the course of systems consolidation. We computed the ERS by contrasting the pattern similarity between the same items during the final run of the encoding task and during the recognition task (encoding-old-similarity, EOS) with the similarity between pattern representations during the final run of the encoding task and corresponding new items on the recognition task (encoding-new-similarity, ENS). To disentangle memory reinstatement, i.e. ERS, from pattern similarity resulting from mere perceptual processes, we focused on group differences in the differential value of EOS vs. ENS. As shown in Fig. 4, we found a significant decrease in hippocampal ERS from 1d to 28d in the PLAC group ($t_{49.90} = 2.66$, $p = 0.010$, $d = 0.74$) while hippocampal ERS significantly increased over time in the YOH group ($t_{46.66} = −2.22$, $p = 0.031$, $d = 0.62$; drug × delay, SVC peak-level: $x = −26$, $y = −10$, $z = −26$, $t = 4.19$, $p(\text{FWE}) = 0.007$, $k = 20$). This finding indicates the expected time-dependent decrease in reinstatement of encoding-related hippocampal pattern representations, implying a decrease in successful retrieval of episodic

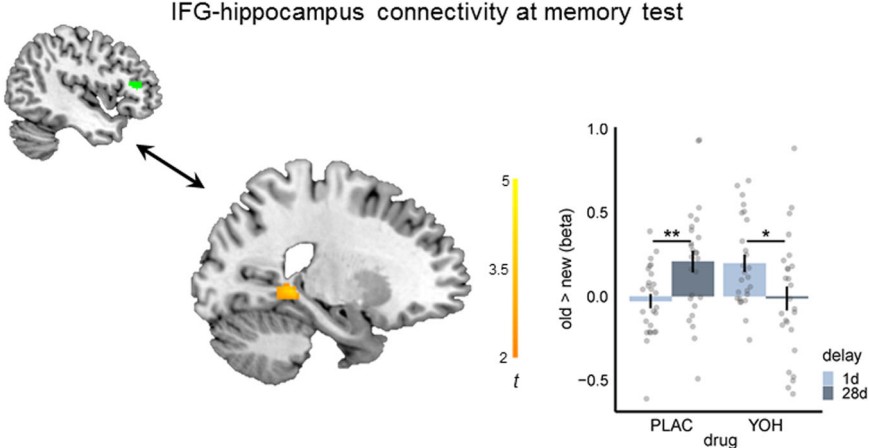

**Fig. 3 Noradrenergic stimulation reverses the time-dependent changes in IFG-hippocampus functional connectivity.** Psychophysiological interaction analysis indicated that while the connectivity between the inferior frontal gyrus (IFG) and hippocampus increased in the placebo (PLAC) group from 28d relative to 1d ($p = 0.004$ two-tailed Welch's $t$-test), there was even a decrease in IFG-hippocampus connectivity at 28d compared to memory testing after 1d in the yohimbine (YOH) group ($p = 0.020$, two-tailed Welch's $t$-test; drug × delay, SVC peak-level: $x = -22$, $y = -40$, $z = -2$, $p(FWE) = 0.009$, mixed ANOVA; $n = 104$ participants). Bonferroni correction was applied for the number of regions of interest in each analysis. The seed-region in the IFG (green), retrieved from the drug × delay interaction of the univariate analysis (peak: $x = -50$, $y = 34$, $z = 12$; $k = 62$), and the significant cluster in the hippocampus (orange) are superimposed on sagittal slices of T1-weighted template images. Distribution of beta-values for the significant cluster is presented for the contrast old > new. Bars represent mean ± SEM. Source data are provided as Source data file. *$p < 0.050$, **$p < 0.010$.

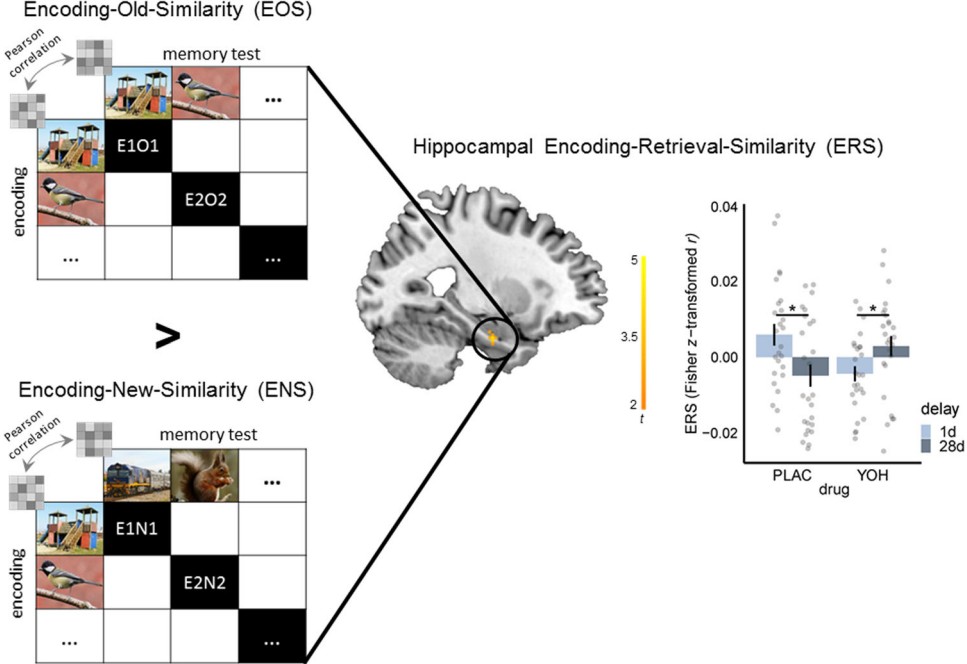

**Fig. 4 Multivariate encoding-retrieval-similarity (ERS) analysis.** Participants of the placebo (PLAC) group showed a significant decrease in hippocampal pattern reinstatement, as reflected in ERS, from 28d relative to 1d ($p = 0.010$, two-tailed Welch's $t$-test), while there was even a significant increase in hippocampal ERS from the 1d to the 28d test in the yohimbine (YOH) group ($p = 0.031$, two-tailed Welch's $t$-test; drug × delay: SVC peak-level: $x = -26$, $y = -10$, $z = -26$, mixed ANOVA; $n = 104$ participants). Bonferroni correction was applied for the number of regions of interest in each analysis. All images are licensed under Creative Commons License; image of the playground courtesy of Tomasz Sienicki (https://commons.wikimedia.org/wiki/File:Playground_29_ubt.JPG; image unchanged); image of the bird courtesy of Francis C. Franklin (https://commons.wikimedia.org/wiki/File:Great_tit_side-on.jpg; image unchanged); image of the train courtesy of DBZ2313 (https://commons.wikimedia.org/wiki/File:Locomotive_NR27_hauling_Indian_Pacific_train_(cropped).jpg; image unchanged); and image of the squirrel courtesy of Peter Timing (https://commons.wikimedia.org/wiki/File:Squirrel_posing.jpg; image unchanged). Visualizations of the ERS results include the $t$-map for drug × delay superimposed on a sagittal section of a T1-weighted template image and the Fisher $z$-transformed $r$-values for the significant cluster in the contrast EOS > ENS. Bars represent mean ± SEM. Source data are provided as Source data file. *$p < 0.050$.

details of individual memories[43] in the PLAC group. The YOH group, in turn, showed even the opposite course with increased similarity between hippocampal patterns representations during encoding with activation patterns at memory testing after 28d vs. 1d, again indicating a reversal in systems consolidation dynamics by post-encoding noradrenergic stimulation.

In addition to the analysis of trial-unique pattern reinstatement, we also analyzed the influence of noradrenergic stimulation on cross-trial ERS, representing general memory-related activity rather than the reinstatement of individual memories. Again, we found a significant decrease in ERS from 1d to 28d for the PLAC group ($t_{49.94} = 2.63$, $p = 0.011$, $d = 0.73$) and a significant time-dependent increase in the YOH group ($t_{47.83} = -2.05$, $p = 0.046$, $d = 0.58$) for the hippocampus (drug × delay, SVC peak-level: x = −24, y = −6, z = −26, t = 3.21, $p$(FWE) = 0.011, k = 22). No significant effects were found for the IFG or vmPFC nor on the whole-brain level in these analyses. The absence of a drug × delay interaction effect on the ERS in neocortical areas might be due to the fact these areas seem to be less involved in the retrieval or reinstatement of specific memory details[32], other than the hippocampus which is thought to play a key role in reconstructing the original memory representation during recall[32] and in coding contextual information such as space and time[44].

**Exploratory analyses of posterior areas**. Given the result of our exploratory whole-brain analysis indicating a time-dependent increase in precuneal activity from encoding to memory testing, which was reversed by noradrenergic stimulation, and due to recent findings indicating an important role of posterior neocortical areas for long-term memory-storage[28,29], we performed additional exploratory analyses including the precuneus, retrosplenial cortex (anatomically defined as Brodmann areas 29 and 30) and the posterior cingulate gyrus representing the posterior parietal cortex (PPC)[45] as well as the angular gyrus. This analysis yielded an interaction of drug × delay for the angular gyrus (SVC peak-level: x = −62, y = −54, z = 22, t = 3.75, $p_{corr}$(FWE) = 0.020, k = 96) with a significant increase in activity from encoding to memory testing in the PLAC group ($t_{45.88} = -2.51$, $p = 0.016$, $d = 0.70$), but—in line with our results in the IFG—a significant decrease in activity in the angular gyrus in the YOH group ($t_{49.82} = 2.41$, $p = 0.020$, $d = 0.67$; see Supplementary Fig. 3A). Apart from this interaction in the angular gyrus and of the above-mentioned effect in the precuneus (drug × delay, SVC peak-level: x = −4, y = −50, z = 46, t = 5.21, $p_{corr}$(FWE) < 0.001, k = 877; see Supplementary Fig. 3B), there were no effects of drug × delay in other PPC-areas. Beyond these changes in precuneal and angular gyral activity from encoding to memory testing, there were no further effects of drug × delay in the tested posterior areas, neither in our univariate or connectivity analyses during memory testing, nor in the multivariate ERS-analyses.

## Discussion

The time-dependent redistribution of memory traces from the hippocampus to neocortical areas, referred to as systems consolidation, has been in the spotlight of memory research for decades[1–6]. Systems consolidation may be highly adaptive in that it aids the building of abstract, generalized knowledge structures but may become detrimental for memories of important events that need to be remembered in detail. Recent evidence suggests that systems consolidation might be more dynamic than initially thought[28,29]. However, whether the systems consolidation process can be experimentally manipulated and shaped by conditions such as emotional arousal remained unclear. Here, we asked whether noradrenergic stimulation shortly after encoding may modulate the systems consolidation process. We show that

pharmacologically enhanced noradrenergic activity shortly after encoding reduces the time-dependent decline of memory performance and, more importantly, increases hippocampal but decreases neocortical involvement in memory from 1d to 28d after encoding. Furthermore, multivariate ERS analysis revealed that while reactivation of hippocampal encoding patterns decreased over time in the PLAC group, after YOH intake there was even a time-dependent increase in the reactivation of hippocampal encoding patterns at the delayed test. Importantly, our autonomic and neuroimaging data indicate that the initial encoding was left unaffected by the drug and groups were comparable in immediate free recall performance, thus confirming that the observed long-term effects were due to altered consolidation processes. Together, these findings show that noradrenergic stimulation during initial consolidation may have long-lasting effects on human memory by reversing time-dependent neural reorganization processes and, therefore, critically challenge our current understanding of systems consolidation dynamics.

Our behavioral data dovetail with previous research demonstrating enhanced memory for emotionally arousing events and a pivotal role of norepinephrine in this emotional memory enhancement[1–6,46]. By explicitly targeting time-dependent changes in memory over an interval of 28d, we show that the decline in memory performance that was observed for neutral stimuli over time was significantly decelerated for emotional stimuli. Interestingly, however, the beneficial effect of YOH on long-term memory performance was comparable for neutral and emotional items suggesting that the impact of post-encoding noradrenergic stimulation is not biased by stimulus-related arousal.

Most importantly, our neural data revealed that noradrenergic stimulation after encoding critically alters the known dynamics of the systems consolidation process. In the PLAC group, hippocampal activity decreased from 1d to 28d after encoding, as predicted by the systems consolidation theory[1–6]. Likewise, ERS, an indicator of episodic memory reinstatement[34–39], decreased significantly in the hippocampus over time in the PLAC group, suggesting that the hippocampal activity patterns became more distinct from the encoding-related patterns as time after encoding proceeded. The decrease in hippocampal involvement in memory was paralleled by a time-dependent increase in the IFG, a region implicated in remote, semantic memory[5,9,10], and this increase in IFG activity was directly correlated with reduced memory performance. Moreover, there was a time-dependent increase in the functional connectivity between IFG and hippocampus in the PLAC group, which has been linked to the generation of semantic associations in previous research[30]. Critically, noradrenergic stimulation after encoding markedly altered all of these time-dependent neural changes. For hippocampal activity, there was not only no decrease but even an increase from 1d to 28d after encoding. Similarly, hippocampal activity patterns during recognition testing resembled the encoding-related patterns even more at the 28d- vs. 1d-delayed test in the YOH group. Conversely, while activity in neocortical areas implicated in semantic memory (i.e. IFG)[9] or long-term storage per se (i.e., precuneus and angular gyrus in exploratory analyses)[28,29] increased over time in the PLAC group, this neocortical activity was even decreased in the 28d- vs. 1d-delayed test in the YOH group. Furthermore, the time-dependent increase in IFG-hippocampus connectivity was not found when participants received YOH before encoding. Together, this pattern of results strikingly mirrors recent findings in rats[22] and indicates that noradrenergic stimulation after encoding may not only decelerate but even reverse systems consolidation and maintain long-term hippocampus-dependent memory performance.

Our results indicate that—other than classically assumed—memories might not necessarily become hippocampus

independent over time but that environmental factors such as post-encoding arousal may actually increase hippocampus dependency over time, in line with the view that the hippocampus might be continuously required for the retrieval of specific encounters[6,7,47]. Our findings further align with a recently proposed neuromodulation theory suggesting that activation of the locus coeruleus-norepinephrine system during post-encoding periods of consolidation amplifies the preferential processing of salient event features of emotional stimuli[46] and the finding that increased post-encoding amygdala-hippocampal-cortical resting state functional connectivity relates to behavioral negative memory bias and the degree of pattern reinstatement after 1d[48]. At the same time, the present findings emphasize the impact of post-encoding noradrenergic arousal on long-term memory, irrespective of valence or arousal of the encoded stimuli.

How may post-encoding noradrenergic stimulation alter systems consolidation? It is well established that the BLA is critically involved in arousal-related changes of memory, which then modulates neuroplasticity processes in memory storage sites such as the hippocampus[49–53]. Direct support for a critical role of the amygdala in the norepinephrine-related modulation of systems consolidation comes from the above-mentioned rodent study suggesting a reversal of systems consolidation, as indicated by opposite changes in DNA methylation and expression of critical memory-associated genes in the hippocampus and neocortex[22]. Specifically, norepinephrine-injection into the BLA shortly after learning was associated with a time-dependent decrease in DNA methylation and increase in transcriptional activation of *Reln* in the hippocampus, compared to saline. As this gene has been shown to increase synaptic plasticity by increasing long-term potentiation[54] and to support the development of synapses in the hippocampus[55] and its demethylation and transcriptional activation has previously been associated with memory formation[56], such epigenetic mechanisms are likely underlying the reversing effect of post-encoding noradrenergic arousal on the course of systems consolidation. Based on these data, it is tempting to speculate that a noradrenergic arousal-related recruitment of the amygdala during initial consolidation may have resulted in a distinct anchoring of memory traces in the hippocampus leading to an increased connectivity between those brain regions, presumably through epigenetically driven transcriptional changes in memory-related genes which may be actively maintained[22]. At the same time, the burst in noradrenergic stimulation might have led to a break between the pass-off of the short-term synaptic consolidation mode in the hippocampus into a systems consolidation mode, keeping memories in the hippocampus. In the present study, we did not find evidence for an involvement of the amygdala in the norepinephrine-driven reversal of systems consolidation. The absence of such evidence, however, might be due to methodological limitations of task-related fMRI. In particular, noradrenergic stimulation was elevated after encoding, when fMRI was not measured any more, and the putative amygdala modulation of memory most likely took place during a loosely defined window of early consolidation that is difficult to target with fMRI. Future studies might use post-encoding resting-state scans to investigate the potential role of the amygdala and its connectivity with the hippocampus or prefrontal areas in norepinephrine-driven changes in early consolidation.

Although YOH administration led to increased hippocampal involvement in memory after 28d, at the 1d interval YOH appeared to be associated with even reduced hippocampal activity compared to PLAC. This pattern of results is also remarkably similar to the above-mentioned findings in rodents indicating reduced hippocampal activity at a short retention interval[22]. Post-encoding noradrenergic stimulation thus seems not only to decelerate but to reverse systems consolidation and the reduced

hippocampal involvement at short delays may be owing to a restructuring that promotes memory maintenance in the long run.

Both, the present study and the antecedent rodent study[22] probed systems consolidation by contrasting recent, i.e. 1d or 2d, respectively, with remote, i.e. 28d old, memories. Although the parallels between the results of these studies are striking, it is important to note that due to the differential lifespan of rodents and humans the temporal dynamics of systems consolidation might differ between species. In both, rodents and humans, the exact time course of systems consolidation is not well understood[57,58]. While we did find a time-dependent memory reorganization from hippocampal to neocortical areas in the PLAC group after 28d, which was reversed by noradrenergic arousal shortly after encoding, this does not necessarily imply that the systems consolidation process was completed at that time point. It has been argued that systems consolidation might continue for months, years or even decades[58]. Thus, although the 28d old memories investigated here may be considered as remote memories, these memories might not be fully consolidated. Future studies are required to determine how post-encoding noradrenergic arousal influences hippocampal and neocortical contributions to remembering at even later stages of the life of a memory. Another possible limitation refers to the modelling of our imaging data based on the item category regardless of the participants' memory responses. This procedure was chosen because of the overall very high memory performance, specifically in our 1d group, resulting in a low number of false alarms and misses. Future studies on the neural basis of time-dependent changes in memory should employ a design that increases the variability in memory performance, for instance by increasing the number of the to-be-encoded stimuli. Furthermore, as prior work on stress and memory has shown quadratic relationships between post-encoding stress hormone administration and subsequent memory[26] and it is generally assumed that arousal exerts quadratic effects on cognitive functions[59], future studies should include different dosages of YOH to further elucidate noradrenergic arousal effects on changes of memory over time.

To conclude, the present study shows that noradrenergic arousal shortly after learning reverses systems consolidation in humans in the sense that it does not only maintain but even increase hippocampal involvement in memory over time and, in parallel, reduces the neocortical contribution and the related time-dependent decline in memory performance. Thus, noradrenergic arousal shortly after encoding does not only prevent the classical systems consolidation process but seems to induce an alternative, reversed consolidation process, in which hippocampal memory involvement is strengthened and neocortical involvement lessened. These findings demonstrate that a fundamental characteristic of memory is much more dynamic than traditionally thought and sensitive to modulation by environmental factors such as arousal. This mechanism could explain the long-term vividness characteristic for memories of emotionally arousing events[16].

## Methods

**Participants and design.** One-hundred-and-nine healthy volunteers (55 males, 54 females, age: M = 24.09 years, SD = 3.92 years) participated in this experiment. Exclusion criteria were checked in a standardized interview and comprised a history of any psychiatric or neurological diseases, medication intake or drug abuse, kidney- and liver-related diseases, body-mass index below 19 or above 26 kg/m², diagnosed cardiovascular problems as well as any contraindications for MRI measurements or YOH intake. Participants were asked to refrain from physical exercise, caffeine, alcohol, and fatty meals within the two hours before the experiment. All participants provided informed consent before taking part in the experiment and received a monetary compensation for participation. The study protocol was approved by the ethics committee of the Medical Chamber Hamburg (PV5480) and was in accordance with the declaration of Helsinki. The Medical

Chamber Hamburg designated this study to be a basic experimental study in humans and it was not designated to be a clinical trial.

Five participants had to be excluded from the analysis because of technical failure ($n = 1$), missing data for day 2 ($n = 1$) or falling asleep during at least one of the MRI sessions ($n = 3$), thus resulting in a final sample of 104 right-handed young adults (52 men and 52 women, age: M = 24.12 years, SD = 3.92 years). This final sample size is in line with other fMRI studies on the effect of stress or stress mediators on memory[24,60] and an a-priori power calculation with G*Power[61] suggested that this sample size is sufficient to detect a medium-sized effect with a power of 0.80.

We used a fully crossed, placebo-controlled, double-blind, between-subjects design with the factors delay (1d vs. 28d) and drug (PLAC vs. YOH) in which participants were pseudo-randomly assigned to one of four groups, each including 13 men and 13 women.

**Experimental procedure.** All testing took place in the afternoon or the early evening (between 1 and 6 pm). After providing informed consent, participants completed the Trier Inventory for the Assessment of Chronic Stress (TICS)[62], the Beck Depression Inventory (BDI-II)[63], and the State-Trait Anxiety Inventory (STAI)[64]. At the beginning of the second experimental day (either 1d or 28d after day 1), participants also filled out the Pittsburgh Sleep Quality Index (PSQI)[65] extended by questions regarding the duration and quality of sleep in the last 24 hours. Groups did not differ in any of these parameters (see Supplementary Results).

Drug administration and manipulation check (day 1): depending on the experimental group, participants received orally either a PLAC or 20 mg YOH, an α2-adrenoceptor antagonist leading to increased noradrenergic stimulation. PLAC and YOH pills were indistinguishable and the experimenter was not aware of participants' group assignment, thus ensuring double-blind testing. The timing and dosage of YOH administration were chosen in accordance with previous studies[23,24] and based on the known pharmacodynamics of YOH showing that a significant drug action can be expected about 60 min after drug intake. We administered the drug immediately before encoding, in order to ensure the action of the drug shortly after encoding, i.e. during initial consolidation. To assess the efficacy of the pharmacological manipulation and the timing of the drug action, we measured systolic and diastolic blood before drug administration (baseline), immediately after encoding and before the free recall task (35 min), immediately after the free recall task (55 min), and another four times every 15 min during a resting phase (70 min, 85 min, 100 min, 115 min after drug administration), in which participants read handed out magazines. Furthermore, we assessed blood pressure before memory testing on day 2 to rule out any group differences in noradrenergic arousal before memory testing.

Memory encoding (day 1): on the first experimental day, participants performed three encoding runs in the MRI scanner. In each run, participants encoded the same 60 stimuli (30 emotionally negative, 30 neutral; for details see supplementary material) presented in random order using MATLAB (The Mathworks, Inc, Natick, US) with the Psychophysics Toolbox extensions[66], i.e., each picture was presented three times across the encoding session. On each trial, a picture was presented for 3 s followed by a jittered fixation period of $4 \pm 1$ s. Participants were instructed to memorize the presented pictures and informed that there will be a subsequent memory test. To make sure that participants remained fully attentive throughout the encoding task, they were instructed to press a button each time the fixation cross appeared. Immediately after the encoding task, participants completed a free recall task outside the MRI. Here, participants had 15 min to name as many stimuli in as much detail as possible, while an experimenter ticked off the correct stimuli from a list.

Memory test (day 2): Depending on the experimental condition, participants returned to the lab either 1d or 28d after day 1. On this second experimental day, participants performed a recognition task in the MRI, which was separated into three consecutive runs. During the memory test, participants saw the 60 pictures that were presented on day 1 (old) and 60 new pictures (as well as additional items that are beyond the scope of the present manuscript and will be reported elsewhere). Each picture was presented for 3 s and participants were requested to indicate via button press whether the shown picture had been presented on day 1 or not using a four-point scale ("definitely new", "rather new", "rather old", "definitely old"). Between trials, a jittered fixation cross was presented for $4 \pm 1$ s. Finally, participants rated, outside the scanner, the arousal and valence of each stimulus shown in the recognition task on two separate 10-point Likert scales (see Supplemental material).

**Analysis of behavioral and physiological data.** Behavioral and physiological data analyses were performed with R version 4.0.2 (https://www.r-project.org/). Blood pressure was analyzed by means of mixed model ANOVAs with the between factors drug (PLAC/YOH) and delay (1d/28d) and the within factor time (baseline or 35 min/55 min/70 min/85 min/100 min/115 min after drug intake). In case of violated sphericity, as indicated by Mauchly's test, Greenhouse-Geisser corrected degrees of freedom and $p$-values are reported.

To control for attentiveness throughout the encoding task, the number of missed responses was analyzed by means of an LMM using the lme4-package[67]

including delay (1d vs. 28d), drug (PLAC vs. YOH) and run (run1 vs. run2/run3) and their interactions as fixed effects and the random intercept of participants. The probability to remember items in the immediate free recall test was analyzed on a single-trial level by means of a binomial generalized LMM. Again, this model included drug (YOH vs. PLAC), delay (1d vs. 28d), emotion (negative vs. neutral) and their interaction as fixed effects as well as the random intercept of participants and stimuli. Analysis of memory performance focused on the sensitivity index $d'$[25]. D'-values were also further analyzed by means of an LMM. This model included drug (YOH vs. PLAC), delay (1d vs. 28d), emotion (negative vs. neutral) and their interactions as fixed effects and the random intercept of participants. All reported $p$-values are two-tailed. Post-hoc $t$-test were applied with Welch's correction. To further investigate whether participants' confidence in recognizing old items differed depending on stimulus emotionality, delay or drug, confidence for hits was analyzed by means of a trial-wise generalized LMM with drug (PLAC vs. YOH), delay (1d vs. 28d), emotion (neutral vs. negative) and their interactions as fixed effects and the random intercept of participants and stimuli. Furthermore, in an additional analysis, we weighted participants' responses by the level of confidence before computing $d'$ and again, analyzed this by means of an LMM with drug (YOH vs. PLAC), delay (1d vs. 28d), emotion (negative vs. neutral) and their interactions as fixed effects and the random intercept of participants. Finally, to rule out potential effects of sex differences on our results, we exploratively analyzed (unweighted) $d'$ by means of an LMM with the factors drug (YOH vs. PLAC), delay (1d vs. 28d), emotion (negative vs. neutral) and sex (female vs. male) and their interactions as fixed effects and the random intercept of participants.

**MRI data acquisition, preprocessing, and analysis.** MRI data were acquired using a 3T Prisma Scanner (Siemens, Germany) with a 64-channel head coil. Each MRI session consisted of three functional runs and a magnetic (B0) field map to unwarp the functional images (TR = 634 ms, $TE_1$ = 4.92 ms, $TE_2$ = 7.38 ms, 40 slices, voxel size = $2.9 \times 2.9 \times 3.0$ mm$^3$, FOV = 224 mm). For the functional scans, T2*-weighted echo planar imaging sequences were used to obtain 2 mm thick transversal slices (TR = 2000ms, TE = 30 ms, flip angle = 60°, FOV = 224). Additionally, a high-resolution T1 weighted anatomical image (TR = 2500 ms, TE = 2.12 ms, 256 slices, voxel size = $0.8 \times 0.8 \times 0.9$ mm$^3$) was collected at the end of the MRI session of day 2.

To allow for magnetic field (T1) equilibration, the first three functional scans were discarded. The images were first realigned and unwarped using the field maps, then coregistered to the structural image followed by a normalization to Montreal Neurological Institute (MNI) space, as implemented in SPM12 (IXI549Space). For the univariate analysis, the images were additionally smoothed with an 8 mm full-width half-maximum Gaussian kernel.

Preprocessing and analysis of the fMRI data was performed using SPM12 (Wellcome Trust Centre for Neuroimaging, London, UK). Multivariate analysis was applied using custom scripts in MATLAB (The Mathworks, Inc, Natick, US). Results of all neural analyses were considered significant at a family-wise error (FWE) corrected threshold of $p < 0.050$. To test our hypotheses, we performed ROI analyses with a-priori defined ROIs using SVC ($p < 0.050$, FWE corrected) with an initial threshold of $p < 0.005$ uncorrected. We corrected for the number of ROIs in the specific analysis by applying Bonferroni correction. In additional exploratory whole-brain analyses, we used an initial significance threshold of $p < 0.050$ FWE-corrected and a 10-voxel extend. The resulting estimates were extracted using the MarsBar Toolbox (http://www.mrc-cbu.cam.ac.uk/Imaging/marsbar.html) to further inspect interaction effects by post-hoc $t$-tests and correlate the neural activity in the relevant ROIs with the sensitivity index $d'$ as a behavioral indicator of memory performance in R.

ROI definition: The anatomical mask for the hippocampus was derived from the Harvard-Oxford subcortical atlas using a probability threshold of 50%. For the IFG and vmPFC, a sphere with 20 mm radius was used that was centered on the peak voxel (x = −50, y = 16, z = 12) derived from 386 imaging studies reporting "IFG" and on the peak voxel (x = −2, y = 46, z = −8) derived from 199 imaging studies reporting "vmPFC", respectively, as determined by meta-analyses conducted on the neurosynth.org platform (status 26/02/2021).

Univariate fMRI analysis: Due to the overall very high memory performance resulting in a low number of misses and false alarms in many participants, we modelled our imaging data based on stimulus category and chose a correlative approach to link these data to behavioral memory performance. On the first level, the functional MRI data were analyzed using general linear modeling (GLM) as implemented in SPM12. For the univariate analysis, the model included one regressor per run and per emotion for the encoding task (6 regressors) and one regressor per emotion and stimulus category for the recognition task (8 regressors) as well as 6 run constants as regressors of no interest. The resulting 20 regressors were convolved with the canonical hemodynamic response function. A high-pass filter of 128 s was used to remove low-frequency drifts and serial correlations in the time series were accounted for using an autoregressive AR(1)-model.

To assess the effect of YOH on the time-dependent change of hippocampal and prefrontal memory dependency, a flexible factorial model (SPMs non-sphericity correction for violation of the i.i.d.-assumption) with the two between-subject factors delay (1d vs. 28d) and drug (YOH vs. PLAC) and the within-subject factor picture type (new vs. old) was applied. As a-priori ROIs, we focused on the hippocampus in the interaction testing for a higher increase for YOH (vs. PLAC)

from 1d to 28d and expected a higher time-dependent increase for PLAC (vs. YOH) in the IFG and vmPFC for old vs. new stimuli.

Additionally, we assessed the influence of group on the increase in BOLD-activity during the same set of stimuli from day 1, specifically the final encoding run, to day 2 by conducting a flexible factorial model with the two between-subject factors delay (1d vs. 28d) and drug (YOH vs. PLAC) and the within factor task (encoding vs. recognition). As a-priori ROIs, we focused on the hippocampus in the interaction testing for a higher time-dependent increase for YOH (vs. PLAC) and expected a higher time-dependent increase for PLAC (vs. YOH) in the IFG and vmPFC for recognition vs. encoding.

Functional Connectivity Analysis: In addition, psychophysiological interaction (PPI) analyses, as implemented in SPM12, were conducted to assess the functional coupling of the hippocampus and the IFG. To this end, the first eigenvariate of the activity time course of the relevant ROI for old pictures and new pictures were extracted and included as seed in the PPI. We used the significant clusters in the hippocampus (peak: x = 22, y = −38, z = 4; k = 10) and the IFG (peak: x = −50, y = 34, z = 12; k = 62) in the interesting interaction of the univariate group-level analysis as seed. A first-level model was set up including the seed, a vector coding the contrast of interest as well as an interaction term, computed as the element by element product of the first two regressors. The resulting interaction contrasts were then analyzed on the second-level to test whether the functional connectivity between hippocampus and IFG differed depending on delay and noradrenergic stimulation and whether the picture was old or new. For the IFG-seed, we used the hippocampus as an a-priori ROI. Using the hippocampus as a seed, a-priori ROIs were the vmPFC and the IFG.

Multivariate Analysis: RSA using a spherical searchlight approach[40–42] was used to assess ERS as a measure of trial-specific episodic reinstatement[34–39]. For this multivariate analysis, each individual trial of the encoding and recognition task was modelled as an individual regressor convolved with a hemodynamic response function along with six session-constants in one GLM per subject using SPM12. No smoothing was performed on the echoplanar imaging data that entered the GLM. To increase the reliability by normalizing for noise[68], the resulting beta-values were further transformed into t-statistics. We then applied a whole-brain searchlight-analysis in which a sphere with a 3-voxel-radius was centered on every voxel of the brain and subjected the resulting set of voxels to an RSA. Please note that our main findings remained largely unaffected when using a 5-voxel radius of the searchlight sphere. We hereby computed the similarity (Pearson's r) between pattern responses during the final run of encoding on experimental day 1 and during old items in the recognition task on day 2 (encoding-old-similarity, EOS) and between pattern responses during the final run of encoding and the corresponding (matched by valence and the occurrence of animals, humans or objects; old and new items were furthermore roughly matched by the first author and an independent rater based on their subjective experience of scene complexity and number of details) new items of the recognition task (encoding-new-similarity, ENS). The resulting r-maps were further Fisher z-transformed and subjected to a flexible factorial model with the two between-subject factors delay (1d vs. 28d) and drug (YOH vs. PLAC) and the within-subject factor similarity (EOS vs. ENS). We further focused on the differential value of EOS vs. ENS, as an indicator of memory reinstatement, i.e. ERS. In addition to trial-specific pattern-reinstatement, we also assessed cross-trial ERS by correlating pattern responses during encoding trials with patterns of all non-corresponding old (EOS) vs. new trials (ENS) of the recognition task. As a-priori-ROIs, we focused on the hippocampus testing for a significantly higher delay-dependent increase in YOH (vs. PLAC) in ERS and expected a significantly higher delay-dependent increase in PLAC (vs. YOH) for the IFG and vmPFC in ERS.

**Reporting summary**. Further information on research design is available in the Nature Research Reporting Summary linked to this article.

## Data availability
The behavioral, autonomic, and fMRI data generated in this study are provided at Github: https://github.com/valentinakrenz/NorSysCons[69]. Source data are provided with this paper.

## Code availability
Custom code used to analyze and model the data is available at Github: https://github.com/valentinakrenz/NorSysCons[69].

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

## Acknowledgements

This study was funded by a grant from the German Research Foundation (DFG) to L.S. and B.R. (SCHW1357/19). We gratefully acknowledge the assistance of Carlo Hiller with the programming of the task and of Vincent Kühn, Miguel Bermudez Alcaide, Anne Tiefert, Anja Turlach, Roberta Souza Lima, Max Emanuel Feucht, and Carolyn Jakubeit during data collection.

## Author contributions

L.S. and B.R. designed research; V.K. performed research; V.K., T.S., and A.A. analyzed data; and V.K., T.S., A.A., B.R., and L.S. wrote the paper.

## Funding

## Competing interests

The authors declare no competing interests.
