## [Peer Review File · Nature Communications]

Noradrenergic arousal after encoding reverses the course of systems consolidation in humansREVIEWER COMMENTS

Reviewer #1 (Remarks to the Author):

Krenz et al. report a very interesting study highlighting a key role for norepinephrine in the modulation of systems consolidation. I particularly appreciated the translational approach followed in assessing if findings reported in rats by Atucha et al. 2017 PNAS (ref 22) would be observed when following a similar experimental logic in humans. The study is sound and elegant, the results clear, and the combination of psychopharmacological and imaging approaches in 109 subjects a clear asset. The manuscript is well written and most of the relevant literature well referenced. Overall, the study represents a novel and important contribution to the understanding of the neurobiology of emotional memories and systems consolidation processes.

I only have the following minor comments that may help improving the manuscript:

- Abstract: Although indicated in the title, it would be useful to the reader if the species would also be mentioned in the abstract.
- Methods: The time of day when each of the experimental phases was performed should be clearly stated.
- Methods: "Shortly after drug administration". Please, indicate how long precisely.
- Results: it would be useful if the experimental design was clearly stated at the start of the Results section. It is only during stats reporting that the reader finds that there are 4 experimental groups. At this point, the reader wonders which ones are those 4 groups.
- Results, page 5, last paragraph. This part is confusing. I was expecting to see the comparison between placebo and yohimbine for immediate free recall test, 2 groups, but the text talks about 4 groups which is not relevant at this point. It would be helpful if an additional analysis could be provided comparing the only 2 meaningful groups at this stage (as the delay of the 2nd testing time is irrelevant at this point).
- Results, page 6, Figure 1C: "The YOH group showed a significantly lower decrease in memory performance from 1d to 28d than the PLAC group." How was this statement confirmed in statistical terms? In the graph, one only sees comparisons between YOH-1d and 28d and between placebo-1d and 28d. In order to indicate a significant difference in the comparison between the 2 observations, levels of statistical significance do not allow this conclusion. Following the significance interaction, a direct comparison between these observations is further required.
- The authors refer to the antecedent rodent study (ref. 22) to this one, in which long-term/remote memory was tested 28 days after training. However, the human lifespan is drastically different to the rodent one. The authors should discuss how can the 28 days elapsed in the current human study relate to the definition of remote memory and systems consolidation in humans. Can we give the term of long-

term memory in the context of systems consolidation in humans to the one evoked in a 28 days post-encoding?

- Recent studies have demonstrated the continuous need of the hippocampus for long-term memory retrieval. It would be relevant to discuss how this finding integrates with the authors' proposed model.

Reviewer #2 (Remarks to the Author):

Krenz et al. show that noradrenergic stimulation after encoding can 'reverse' systems memory consolidation processes in the sense that hippocampal activity increases over an interval of 28 days whereas neocortical activity decreases over the same interval. A recent study in rodents has shown a similar effect. These results are really striking, particularly because this effect occurs robustly in both rodents and humans. The authors use a well-controlled experimental manipulation which allows attributing the changes in systems consolidation patterns to yohimbine intake. Furthermore, they performed rigorous analyses that test different angles of the memory consolidation processes. The coherence of their results across diverse behavioral and neural measures of systems consolidation is impressive. The current findings are highly relevant to the field: the authors show that the course of memory formation does not progress as rigidly as previously thought, but can instead be flexibly modulated: for example by factors like emotional arousal. I strongly support publication of these findings! The paper is very well written, congratulations to the authors on these interesting findings!

Still, there are a few points I would like the authors to consider:

1. When reading "reversal" of systems consolidation, I envision a systems consolidated trace that is later reset to its original hippocampal dependent state (e.g. as in Winocur Learn Mem 2009, memory reconsolidation). This is not what the authors show here! On the contrary, a first traditional systems consolidation seems to be prevented. On the contrary, the authors show an altered form of "systems" consolidation, where neocortical dependence lessens over time whereas hippocampal contributions are strengthened. To avoid confusion due to terminology, maybe a different phrasing could be used to describe the results?

2. As ROIs, the authors chose the hippocampus as the traditional early site of memory storage, and the IFG and vmPFC as a site for systems consolidated memory, because they have been implicated in semantic memory. Moreover, a previous study in rodents has investigated a prefrontal region as a potential locus for remote memory. However, the stimulus material and the memory tests used do not specifically test for semantization of the learning material. Therefore, it would be reasonable to analyze ROIs that haven been associated with long-term memory storage more generally (e.g. posterior parietal cortex, angular gyrus). Do these regions show similar effects as the prefrontal cortex? The precuneus,

e.g., is reported to show a similar pattern of results as the IFG in a later whole brain analysis. If other aspects of the findings don't show up in these regions, could this mean that specifically frontal regions show a "reversal" of systems consolidation, whereas regions in the perceptual streams show mnemonic activity that is not specifically impacted by drug administration?

3. Pp. 7-8: Did the authors perform a whole-brain analysis for the analysis reported on pp. 7-8? This would be reasonable to perform and report (e.g. in the supplement). I understand that statistical thresholds for whole brain analyses are more stringent than for small volume correction, and I am absolutely aware that the specification of ROIs in this paper was based on the previous literature in the field. I do not think that a failure to observe a whole brain affect in the IFG or hippocampus should alter the presentation or interpretation of the current results. If other regions show the expected pattern of results, the authors may want to consider discussing, however (following a similar rationale as what was done for the analyses reported on pp. 9-10).

4. Pp. 9-10: The rationale for the analysis of drug x delay for recognition (day1 day28) vs encoding (run3) is not as clearly laid out as other analyses reported in the paper. Could the authors add a sentence or two on what exactly was compared and why? Or restructure the paragraph? That would make the reading experience even smoother. In the current state, I had to double back and look at the figure to understand what was done exactly.

5. Pp. 9-10: Following an exploratory whole-brain analysis, the authors report a similar pattern of results for the precuneus as for the IFG (pp. 9-10, increases in activity from encoding to the delayed retrieval in plac, decreases in yoh). This result is well in line with Brodt et al. (2016, 2018), who specifically investigated systems consolidation of declarative material and identified the precuneus as a site for long-term memory storage.

6. Pp. 10-11: Adding to the previous point(s), I would suggest performing the connectivity analysis during the recognition task also for the precuneus and other neocortical regions that may show a significant contribution in the analyses suggested under points 2 and 3, and not only for the IFG (pp. 9-10).

7. P. 23: The searchlight for the RSA uses a 3-voxel-radius (27 voxels). Multivariate analyses leverage the possibility to pinpoint effects across a large number of features. Performing the RSA analyses following the same ROI based approach as applied throughout the paper would, in my view, allow the strongest claims. If this proves difficult, I would consider to repeat the searchlight with an at least 5-voxel-radius (125 voxels) to increase the RSA's "sample size" and avoid false positives. This suggestion is for rigor only. All results tie in so nicely in this paper, that I would not make publication conditional on these results.

8. Pp. 11-12: It would be great if the authors explained the rationale behind an increased ERS and its meaning for memory reinstatement and memory-related activity more clearly in the paper, because this analysis might be hard to understand for readers who are not familiar with these methods.

9. Pp. 11-12: Why do the authors think that a reversal pattern for neocortical regions could not be observed in the RSA?

10. P. 20: Which version of MNI space was used exactly (2006, 2009)?

Reviewer #3 (Remarks to the Author):

Krenz and colleagues investigated the impact of post-encoding noradrenergic stimulation on systems consolidation in humans. The study is of immediate interest to the field of memory and offers novel evidence that systems consolidation can not only be slowed, but reversed. Using complementary univariate and multivariate fMRI analysis approaches, the results revealed time-dependent increases of neural patterns of episodic reinstatement in the hippocampus and time-dependent decreases in neocortical activity for the YOH group from 1d to 28 d—evidence for reversal of systems consolidation that might maintain memory vividness over time. The statistical approaches are valid and appropriate and there is ample information to reproduce the work. I have a few comments below that, if addressed, could help improve the impact of the paper.

Major:

The authors draw several conclusions about emotional modulation of memory, but their behavioral and neuro-imaging findings focus on effects collapsed across valence. For instance, although the results replicate slower forgetting of emotional stimuli, the memory decline from 1d to 28d was weaker in the YOH group than in the PLAC group irrespective of the emotionality of the encoded stimuli. This reviewer thinks that the concluding sentence of the abstract (“potentially explaining the long-term vividness of emotionally arousing memories”) is a bit of a stretch based on the current data that showed effects regardless of stimulus emotionality. Yonelinas and Ritchey (2015) discuss how post-encoding stress does

not preferentially slow forgetting of emotional items suggesting post-encoding adrenergic stimulation would influence memory overall, consistent with the present findings.

It appears the fMRI models assessed old and new stimuli based on objective study history, regardless of the participants' memory responses. Are hits and misses analyzed together? False alarms and correct rejections for new stimuli? If so, the authors might consider an explanation and highlight a limitation that some trials included in the analyses might reflect implicit memory processes or false alarms. Memory performance is very high in this study, but if the hit rate was 85% on average, that means that ~10 forgotten images and 2-3 false alarms were analyzed on average in the fMRI models. It would be cleaner to model out misses and false alarms separately from hits and misses on the recognition test in the fMRI models, or clarify/justify this approach.

Further regarding the emotion condition: It is known that the time-dependent emotion advantage tends to be limited to recollection vs. familiarity. Did the authors consider using their confidence data to explore the possibility of the three-way interaction on their higher confidence images? (i.e., high confidence d' based on high confidence hits and fas). The three way interaction of d' appears to be trending (drug \times delay \times emotion: $\beta=0.31$, $p=0.127$) and could be influenced by confidence. Why did the authors not leverage the confidence ratings, at least in a supplementary behavioral analysis?

This paper is particularly timely, given the emotion and reward modulated memory field has become increasingly interested in post-encoding MTL-neocortical consolidation effects, linking to the literature in the discussion might help improve the impact of the field and connect with more of the recent proposed theoretical frameworks. For instance, Clewett and Murty's (2019) influential neuromodulation theory purports that activation of the LC-NE system during post-encoding periods of consolidation will also amplify the preferential processing of salient event features of emotional stimuli even further, in part through amygdala modulation. Further, Kark and Kensinger have shown increased post-encoding amygdala-hippocampal-occipital resting state functional connectivity relates to behavioral negative memory bias and the degree of occipital reinstatement at 24 hr retrieval. Murty and colleagues (2017) showed that post-encoding mesolimbic-cortical connectivity predicts high reward memory 24 hours after encoding (Murty et al., 2017). What do the authors propose is happening during these post-encoding periods when NA is administered to disrupt these processes (is there a break between the pass-off of the short term synaptic consolidation mode into to systems consolidation mode, keeping memories in the MTL?).

In their discussion the authors touch on the reduced hippocampal involvement at short delays and state that it may be 'owing to a restructuring that promotes memory maintenance in the long run'. How might reduced (below 0) hippocampal ERS in the YOH group after a short delay facilitate a later boost in hip ERS at 28 days? How and why does NA stimulation increase IFG-hippo FC after 1 day (above 0) in the YOH group? More discussion on the possible reasons and mechanisms of this surprising reversal, beyond slowing, would be helpful.

Minor:

Prior work on stress and memory has shown quadratic relationships between post-encoding stress hormone admin and subsequent memory as well as sex differences (Andreano and Cahill, 2006), did the authors consider quadratic relationships? A brief explanation for focusing on the linear effects of stress would be helpful in the paper. Although perhaps underpowered, a mention of the possibility of sex-differences in the limitations section would be helpful.

Images were matched on visual complexity, what kind of visual complexity metric was quantified?

References

Andreano, J.M. and Cahill, L. (2006) Glucocorticoid release and memory consolidation in men and women. *Psychol. Sci.* 17, 466–470

Clewett, D., and V. P. Murty. 2019. "Echoes of Emotions Past: How Neuromodulators Determine What We Recollect." *eNeuro* 6 (2). <https://doi.org/10.1523/eneuro.0108-18.2019>.

Kark, Sarah M., and Elizabeth A. Kensinger. 2019. "Post-Encoding Amygdala-Visuosensory Coupling Is Associated with Negative Memory Bias in Healthy Young Adults." *The Journal of Neuroscience: The Official Journal of the Society for Neuroscience* 39 (16): 3130–43.

MurtyVP, TomparyA, AdcockRA, DavachiL (2017) Selectivity in postencoding connectivity with high-level visual cortex is associated with reward-motivated memory. *J Neurosci* 37:537–545.

Yonelinas, Andrew P., and Maureen Ritchey. 2015. "The Slow Forgetting of Emotional Episodic Memories: An Emotional Binding Account." *Trends in Cognitive Sciences* 19 (5): 259–67.

Responses to reviewers

Reviewer #1

We thank the reviewer for his/her positive and constructive comments. We are glad that he/she considers our study novel, important, sound and elegant, our results clear, the combination of psychopharmacological and imaging approaches in 109 subjects a clear asset and our manuscript well written.

- *Abstract: Although indicated in the title, it would be useful to the reader if the species would also be mentioned in the abstract.*

RESPONSE: We agree and have added the examined species (humans) to the abstract. Please see page 2, lines 26 to 27:

„Here, we show in healthy humans that the dynamics of systems consolidation can be experimentally manipulated and even reversed.“

- *Methods: The time of day when each of the experimental phases was performed should be clearly stated.*

RESPONSE: All testing phases (day 1 and day 2) took place in the afternoon or the early evening. In response to this comment, we have specified the exact time of testing in the section reporting our experimental procedure. Please see page 21, line 539:

„All testing took place in the afternoon or the early evening (between 1 and 6pm).“

- *Methods: “Shortly after drug administration”. Please, indicate how long precisely.*

RESPONSE: In response to this comment we specified how long after drug administration encoding took place. Please see page 5, lines 120 to 122:

„ Within 5min after drug administration, participants encoded 60 pictures (30 neutral, 30 emotionally negative) in the MRI scanner, each presented once in each of three consecutive runs.“

- *Results: it would be useful if the experimental design was clearly stated at the start of the Results section. It is only during stats reporting that the reader finds that there are 4 experimental groups. At this point, the reader wonders which ones are those 4 groups.*

RESPONSE: In response to this comment, we added a sentence specifying the experimental design at the beginning of our results section. Please see page 4, lines 98 to 101:

„To determine the effect of post-encoding noradrenergic arousal on time-dependent systems consolidation in humans, we used a two-day fully crossed between-subjects design with the factors drug (PLAC vs. YOH) and delay (1d vs. 28d), resulting in four experimental groups: 1d/PLAC, 28d/PLAC, 1d/YOH and 28d/YOH.“

- *Results, page 5, last paragraph. This part is confusing. I was expecting to see the comparison between placebo and yohimbine for immediate free recall test, 2 groups, but the text talks about 4 groups which is not relevant at this point. It would be helpful if an additional analysis could be provided comparing the only 2 meaningful groups at this stage (as the delay of the 2nd testing time is irrelevant at this point).*

RESPONSE: We agree that the delay of the 2nd testing time is theoretically irrelevant at this point. However, we still subjected the immediate free recall data to a generalized linear mixed model (LMM) including the factors emotion, drug and delay because the mere fact that the delay was not relevant at this time point does not exclude the possibility that the 1d and 28d groups did differ for some reason in initial encoding, which would have important implications for our interpretation of day 2 memory performance. Thus, we do think that the data showing that the four groups did not differ in initial encoding is important and should be explicitly reported. Nevertheless, we agree that the comparison of the PLAC and YOH groups is particularly important. This comparison, however, is included in the model we run; it is reflected in the factor drug. Our results showed that there was no main effect of drug ($p=0.499$), thus indicating that the drug groups did not differ in memory encoding. An additional *t*-test comparing the PLAC and YOH groups would be redundant, in our view. However, in response to this comment, we made our rationale for running a model including the factor delay clearer. Please see pages 5 to 6, lines 130 to 135:

„Although the delay to the recognition test should not be relevant for performance immediately after encoding, we ran a trial-wise binomial generalized linear mixed model (LMM) with drug (PLAC vs. YOH), delay (1d vs. 28d) and emotion (neutral vs. negative) and their interactions as fixed effects and the random intercept of participants and stimuli to not only assess potential drug effects on encoding but to also rule out potential differences between the 1d- and 28d-groups in initial encoding.“

- *Results, page 6, Figure 1C: “The YOH group showed a significantly lower decrease in memory performance from 1d to 28d than the PLAC group.” How was this statement confirmed in statistical terms? In the graph, one only sees comparisons between YOH-1d and 28d and between placebo-1d and 28d. In order to indicate a significant difference in the comparison between the 2 observations, levels of statistical significance do not allow this conclusion. Following the significance interaction, a direct comparison between these observations is further required.*

RESPONSE: Please note that we intentionally did not use standard ANOVA models but LMMs. One of the many advantages of using regression analyses, in this case multilevel regression models, is that these allow inferences about the directionality of effects based on the sign of the resulting estimates¹. Therefore, in this case – differently to ANOVAs – no post-hoc tests are needed to interpret the results. For instance, because of the dummy-coding of our factor delay with 0=1d vs. 1=28d, the main effect delay with a β -estimate of -1.12 (95%-CI[-1.12,-0.72], $t_{125.82}=-5.43$, $p<0.001$) indicates that d' decreases with increasing delay by -1.12 while controlling for the other factors at reference level, i.e. drug=PLAC and emotion=neutral. Because of the dummy-coding of the factor drug with 0=PLAC vs. 1=YOH, the interaction drug \times delay with an β -estimate of 0.64 (95%-CI[0.06, 1.21], $t_{125.82}=2.18$, $p=0.029$) indicates that the YOH group decreases by 0.64 less in d' (compared to PLAC) from 1d to 28d while controlling for the factor emotion. Therefore, while d' decreases from 1d to 28d for both the PLAC and YOH groups, this time-dependent decrease was significantly lower in the YOH group than in the PLAC Group, as shown in Fig. 1C. However, we agree that by looking at the figure and its caption alone without the inclusion of any β -estimates, it is difficult to interpret the direction of the drug \times delay interaction. Further, the asterisks might be misleading as they were more for illustrative purposes and did not reflect the critical statistical test. We now removed these asterisks and left only the indication of the significant interaction in order to avoid confusion. We further rephrased the caption of Fig. 1C to include the required β -estimates. Please see page 7, lines 153 to 156:

„However, while memory performance significantly decreased from 1d to 28d after encoding (main effect delay: $\beta=-1.12$, $p<0.001$), post-encoding noradrenergic arousal reduced this time-dependent memory decline (drug \times delay: $\beta=0.64$, $p=0.029$): The YOH group showed a significantly smaller decrease in memory performance from 1d to 28d than the PLAC group.“

- *The authors refer to the antecedent rodent study (ref. 22) to this one, in which long-term/remote memory was tested 28 days after training. However, the human lifespan is drastically different to the rodent one. The authors should discuss how can the 28 days elapsed in the current human study relate to the definition of remote memory and systems consolidation in humans. Can we*

give the term of long-term memory in the context of systems consolidation in humans to the one evoked in a 28 days post-encoding?

RESPONSE: We agree that due to the shorter life span it is likely that rodent's systems consolidation might proceed faster than in humans. However, when and for how long memory reorganization in the course of systems consolidation occurs remains largely unclear for both species^{2,3}. While it has been assumed that human systems consolidation needs weeks, months, years or even the life time of a memory to be accomplished², changes at the brain systems level are evident after only one night of sleep⁴. Moreover, our findings of an increased neocortical and decreased hippocampal activation and pattern reinstatement during memory testing in PLAC are in line with a systems consolidation process after 28d compared to 1d and we therefore think that it is justified to refer to the 28d-old memory as *remote memory*. However, we do agree that the time-dependent reorganization may not have been finished after 28d. We discuss this aspect and potential species differences now on page 19, lines 473 to 486:

„Both, the present study and the antecedent rodent study²² probed systems consolidation by contrasting recent, i.e. 1d or 2d, respectively, with remote, i.e. 28d old, memories. Although the parallels between the results of these studies are striking, it is important to note that due to the differential lifespan of rodents and humans the temporal dynamics of systems consolidation might differ between species. In both, rodents and humans, the exact time course of systems consolidation is not well understood^{57,58}. While we did find a time-dependent memory reorganization from hippocampal to neocortical areas in the PLAC group after 28d, which was reversed by noradrenergic arousal shortly after encoding, this does not necessarily imply that the systems consolidation process was completed at that time point. It has been argued that systems consolidation might continue for months, years or even decades⁵⁸. Thus, although the 28d old memories investigated here may be considered as remote memories, these memories might not be fully consolidated. Future studies are required to determine how post-encoding noradrenergic arousal influences hippocampal and neocortical contributions to remembering at even later stages of the life of a memory.“

- *Recent studies have demonstrated the continuous need of the hippocampus for long-term memory retrieval. It would be relevant to discuss how this finding integrates with the authors' proposed model.*

RESPONSE: While we did find a clear decrease in hippocampal involvement during memory testing in our PLAC group, the findings of our YOH group show that memories might not necessarily become hippocampus independent over time, but under the influence of environmental factors such as arousal become even more hippocampus dependent over the time course of at least 28d. This

finding is actually well in line with recent studies showing that the hippocampus might be required even for remote memory. We now discuss this issue on page 17, lines 425 to 429:

„Our results indicate that – other than classically assumed – memories might not necessarily become hippocampus independent over time but that environmental factors such as post-encoding arousal may actually increase hippocampus dependency over time, in line with the view that the hippocampus might be continuously required for the retrieval of specific encounters^{6,7,47}.“

Reviewer #2

We thank the reviewer for his/her encouraging and very constructive comments. We are glad that he/she considers our experimental manipulation well-controlled, our manuscript well-written, our analyses rigorous and our findings impressive and highly relevant to the field.

1. *When reading “reversal” of systems consolidation, I envision a systems consolidated trace that is later reset to its original hippocampal dependent state (e.g. as in Winocur Learn Mem 2009, memory reconsolidation). This is not what the authors show here! On the contrary, a first traditional systems consolidation seems to be prevented. On the contrary, the authors show an altered form of “systems” consolidation, where neocortical dependence lessens over time whereas hippocampal contributions are strengthened. To avoid confusion due to terminology, maybe a different phrasing could be used to describe the results?*

RESPONSE: We see the reviewer’s point. However, in our view our pattern of results may indeed be described as a reversed course of systems consolidation because while systems consolidation is classically assumed to include a decrease in hippocampal and increase in neocortical involvement in memory, we show here, after noradrenergic stimulation, the reversed pattern with increased hippocampal and decreased neocortical involvement over time. We thought about an alternative verb but did not find one that would be equally well suited to capture our pattern of results. The potential for confusion, however, might come primarily from the ambiguity related to when the process is reversed, i.e. either early before the classical systems consolidation occurred (which is what we see here) or late after it had already occurred. To avoid confusion, we make it now more explicit that we are referring to a reversal of the systems consolidation process immediately after encoding, i.e. before the classical process had occurred. We have changed the title of our manuscript accordingly. It reads now:

„Noradrenergic arousal after encoding reverses the course of systems consolidation in humans“

Moreover, we now indicate more explicitly what we mean with the reversal of systems consolidation by noradrenergic arousal. Please see page 20, lines 497 to 504:

„To conclude, the present study shows for the first time that noradrenergic arousal shortly after learning reverses systems consolidation in humans in the sense that it does not only maintain but even increase hippocampal involvement in memory over time and, in parallel, reduces the neocortical contribution and the related time-dependent decline in memory performance. Thus, noradrenergic arousal shortly after encoding does not only prevent the classical systems consolidation process but seems to induce an alternative, reversed consolidation process, in which hippocampal memory involvement is strengthened and neocortical involvement lessened.“

2. *As ROIs, the authors chose the hippocampus as the traditional early site of memory storage, and the IFG and vmPFC as a site for systems consolidated memory, because they have been implicated in semantic memory. Moreover, a previous study in rodents has investigated a prefrontal region as a potential locus for remote memory. However, the stimulus material and the memory tests used do not specifically test for semantization of the learning material. Therefore, it would be reasonable to analyze ROIs that haven been associated with long-term memory storage*

more generally (e.g. posterior parietal cortex, angular gyrus). Do these regions show similar effects as the prefrontal cortex? The precuneus, e.g., is reported to show a similar pattern of results as the IFG in a later whole brain analysis. If other aspects of the findings don't show up in these regions, could this mean that specifically frontal regions show a "reversal" of systems consolidation, whereas regions in the perceptual streams show mnemonic activity that is not specifically impacted by drug administration?

RESPONSE: In response to this comment, we applied additional exploratory ROI-analyses focusing on the angular gyrus and the PPC (i.e. precuneus, retrosplenial cortex, posterior cingulate cortex)⁵. We report the results of these additional, exploratory analyses on page 15, lines 351 to 368:

„Given the result of our exploratory whole-brain analysis indicating a time-dependent increase in precuneal activity from encoding to memory testing, which was reversed by noradrenergic stimulation, and due to recent findings indicating an important role of posterior neocortical areas for long-term memory-storage^{28,29}, we performed additional exploratory analyses including the precuneus, retrosplenial cortex (anatomically defined as Brodmann areas 29 and 30) and the posterior cingulate gyrus representing the posterior parietal cortex (PPC)⁴⁵ as well as the angular gyrus. This analysis yielded an interaction of drug×delay for the angular gyrus (SVC peak-level: x=-62, y=-54, z=22, t=3.75, $p_{corr}(FWE)=0.020$, k=96) with a significant increase in activity from encoding to memory testing in the PLAC group ($t_{48.85}=-3.86$, $p<0.001$, $d=1.07$), but – in line with our results in the IFG – a significant decrease in activity in the angular gyrus in the YOH group ($t_{47.95}=2.27$, $p=0.028$, $d=0.67$; see supplementary figure 3A). Apart from this interaction in the angular gyrus and of the above-mentioned effect in the precuneus (drug×delay, SVC peak-level: x=-4, y=-50, z=46, t=5.21, $p_{corr}(FWE)<0.001$, k=877; see supplementary figure 3B), there were no effects of drug×delay in other PPC-areas. Beyond these changes in precuneal and angular gyral activity from encoding to memory testing, there were no further effects of drug×delay in the tested posterior areas, neither in our univariate or connectivity analyses during memory testing, nor in the multivariate ERS-analyses.“

Moreover, we address these findings briefly but make also clear that these findings stem from an additional, exploratory analysis and thus need to be confirmed by future studies. Please see page 17, lines 416 to 419:

„Conversely, while activity in neocortical areas implicated in semantic memory (i.e. IFG)⁹ or long-term storage per se (i.e. precuneus and angular gyrus in exploratory analyses)^{28,29} increased over time in the PLAC group, this neocortical activity was even decreased in the 28d- vs. 1d-delayed test in the YOH group.“

3. *Pp. 7-8: Did the authors perform a whole-brain analysis for the analysis reported on pp. 7-8? This would be reasonable to perform and report (e.g. in the supplement). I understand that statistical thresholds for whole brain analyses are more stringent than for small volume correction, and I am absolutely aware that the specification of ROIs in this paper was based on the previous literature in the field. I do not think that a failure to observe a whole brain affect in the IFG or hippocampus should alter the presentation or interpretation of the current results. If other regions show the expected pattern of results, the authors may want to consider discussing, however (following a similar rationale as what was done for the analyses reported on pp. 9-10).*

RESPONSE: In addition to our a-priori ROI analyses, we applied exploratory whole-brain analyses to all of our imaging analyses, and reported significant findings in case there were any (as indicated on page 25, lines 636 to 638). To make this clearer we now also included the lack of whole-brain-findings on page 9, lines 228 to 229:

„There were no effects of drug×delay in the vmPFC or in an exploratory whole-brain analysis.”

4. *Pp. 9-10: The rationale for the analysis of drug x delay for recognition (day1 day28) vs encoding (run3) is not as clearly laid out as other analyses reported in the paper. Could the authors add a sentence or two on what exactly was compared and why? Or restructure the paragraph? That would make the reading experience even smoother. In the current state, I had to double back and look at the figure to understand what was done exactly.*

RESPONSE: In response to this comment, we restructured the introductory part of this section and hope that the rationale behind this analysis is now easier to understand. Please see pages 10 to 11, lines 249 to 253:

„While the previous analysis focused on brain activity for old vs. new items during memory testing, in a next step we analyzed changes in brain activity from the last run of encoding to recognition testing either 1d or 28d later. By taking explicitly the activity at encoding into account, this analysis provides insights into dynamic changes in memory-related activity over time and its modulation by noradrenergic arousal.“

5. *Pp. 9-10: Following an exploratory whole-brain analysis, the authors report a similar pattern of results for the precuneus as for the IFG (pp. 9-10, increases in activity from encoding to the delayed retrieval in plac, decreases in yoh). This result is well in line with Brodt et al. (2016, 2018), who specifically investigated systems consolidation of declarative material and identified the precuneus as a site for long-term memory storage.*

RESPONSE: We now included a reference to the exciting work of Brodt et al.^{6,7} when reporting our exploratory whole-brain result for the precuneus. Please see page 11, lines 261 to 267:

„Interestingly, an exploratory whole-brain analysis also revealed a significant drug×delay interaction for recognition vs. encoding in the same direction for the precuneus (drug×delay, peak-level: $x=-4$, $y=50$, $z=46$, $t=5.21$, $p(\text{FWE})=0.009$, $k=27$), showing a significant increase in activity after a short compared to a long retention delay in the PLAC group ($t_{48.69}=-3.56$, $p<0.001$, $d=0.99$), in line with recent findings that identified the precuneus as a site for neocortical long-term storage^{28,29}, but a significant decrease in activity for the YOH group ($t_{47.12}=3.95$, $p<0.001$, $d=1.10$).“

6. *Pp. 10-11: Adding to the previous point(s), I would suggest performing the connectivity analysis during the recognition task also for the precuneus and other neocortical regions that may show a significant contribution in the analyses suggested under points 2 and 3, and not only for the IFG (pp. 9-10).*

RESPONSE: As suggested, we performed the connectivity analyses also for the precuneus and other parietal areas. These analyses, however, yielded no significant findings, which has been added to the text on page 15, lines 365 to 368:

„Beyond these changes in precuneal and angular gyral activity from encoding to memory testing, there were no further effects of drug×delay in the tested posterior areas, neither in our univariate or connectivity analyses during memory testing, nor in the multivariate ERS-analyses.”

7. P. 23: *The searchlight for the RSA uses a 3-voxel-radius (27 voxels). Multivariate analyses leverage the possibility to pinpoint effects across a large number of features. Performing the RSA analyses following the same ROI based approach as applied throughout the paper would, in my view, allow the strongest claims. If this proves difficult, I would consider to repeat the searchlight with an at least 5-voxel-radius (125 voxels) to increase the RSA's "sample size" and avoid false positives. This suggestion is for rigor only. All results tie in so nicely in this paper, that I would not make publication conditional on these results.*

RESPONSE: We chose a searchlight-based RSA-approach, because it allows both an analysis of our a-priori defined ROIs as well as an exploratory whole-brain analysis based on flexible factorial models as implemented in SPM12 and therefore aligns nicely with the second level analysis rationale throughout the whole manuscript while expanding our findings on the univariate level to multivariate pattern similarities between encoding and retrieval, i.e. episodic memory reinstatement. We would therefore prefer to stick to this searchlight-based approach.

In response to this reviewer's comment, we did repeat our trial-unique ERS-analysis using a searchlight-radius of 5 voxels. As can be seen in the figure below, we again found a significant drug \times delay interaction for the hippocampus (SVC peak-level: $x=-32, y=-12, z=-20, t=3.55, p(\text{FWE})=0.030, k=18$), indicating a decrease in hippocampal pattern reinstatement over time for PLAC ($t_{46.70}=2.01, p=0.050$) while ERS tended to increase over time for YOH ($t_{50.00}=-1.99, p=0.053$).

Again, we did not find an interaction of drug \times delay in any other a-priori ROIs (i.e. vmPFC and IFG) nor in the exploratory posterior ROIs (suggested by this reviewer i.e. angular gyrus and PCC) indicating that the absence of effects of delay and noradrenergic arousal on neocortical ERS did not depend on the size of our searchlight sphere. Thus, these analyses show that our findings appear to be relatively robust against variations in the radius of the searchlight sphere, which is reassuring.

However, due to the autocorrelation of adjacent voxels, an increase of the searchlight radius does not necessarily increase the sample size of the analysis while increasing the number of voxels taken into account in mapping the encoding-retrieval pattern similarity into the central searchlight voxel of the resulting r -maps. Given the small size and longitudinal shape of the hippocampus, which has been in the focus of the research of systems consolidation and hence this study, an increased searchlight sphere results in the consideration of activation patterns of neighboring areas into the hippocampal ERS and therefore decreases the sensitivity of this analysis. Furthermore, our analytic strategy is close to the recommendation by Kriegeskorte et al.⁸ which also suggests a different radius of the searchlight sphere depending on the size of the area of interest.

We would therefore prefer to stick to the reported results using a 3-voxel radius. However, we indicate now explicitly in the text that our main results were largely unaffected to variations in the radius of the searchlight sphere. Please see page 27, lines 693 to 694:

„Please note that our main findings remained largely unaffected when using a 5-voxel radius of the searchlight sphere.“

8. *Pp. 11-12: It would be great if the authors explained the rationale behind an increased ERS and its meaning for memory reinstatement and memory-related activity more clearly in the paper, because this analysis might be hard to understand for readers who are not familiar with these methods.*

RESPONSE: We agree and have now explained the meaning of an altered ERS in more detail. Please see page 13, lines 307 to 310:

„Because a decrease in memory reinstatement is thought to reflect a more abstract memory representation and that the episodic details of a specific memory are not successfully retrieved⁴³, we expected a decrease in similarity between activation patterns during encoding and memory testing, i.e. ERS, in the course of systems consolidation.“

As well as on page 13, lines 319 to 325:

„This finding indicates the expected time-dependent decrease in reinstatement of encoding-related hippocampal pattern representations, implying a decrease in successful retrieval of episodic details of individual memories⁴³ in the PLAC group. The YOH group, in turn, showed even the opposite course with increased similarity between hippocampal patterns representations during encoding with activation patterns at memory testing after 28d vs. 1d, again indicating a reversal in systems consolidation dynamics by post-encoding noradrenergic stimulation.“

9. *Pp. 11-12: Why do the authors think that a reversal pattern for neocortical regions could not be observed in the RSA?*

RESPONSE: Other than the univariate analysis, which focusses on differences between our groups on activation-level contrasts between image categories, the ERS-analysis focusses on the similarity of activation patterns between individual items during encoding and memory test, i.e. episodic memory reinstatement. The fact that we found an ERS effect, which was influenced by retention delay and noradrenergic stimulation, for the hippocampus is consistent with its role in reconstructing the original memory representation during memory test due to its function in pattern completion⁹ and in coding contextual information such as space and time¹⁰. Neocortical areas, however, are considered to be less relevant for retrieving specific memory-details⁹. We have added this potential explanation to the text. Please see page 14, lines 333 to 337:

„ The absence of a drug×delay interaction effect on the ERS in neocortical areas might be due to the fact these areas seem to be less involved in the retrieval or reinstatement of specific memory details³², other than the hippocampus which is thought to play a key role in reconstructing the original memory representation during recall³² and in coding contextual information such as space and time⁴⁴.“

10. P. 20: *Which version of MNI space was used exactly (2006, 2009)?*

RESPONSE: We used the MNI space IXI549Space as implemented in SPM12. This reference space is based on scans acquired within the IXI dataset (<https://brain-development.org/>), which had been linearly transformed to ICBM MNI 452. Note that MNI152 NLIN 2009 has not yet been adopted by SPM12. We have now added the specification of the MNI space to our methods section on page 25, lines 626 to 628:

„The images were first realigned and unwarped using the field maps, then coregistered to the structural image followed by a normalization to Montreal Neurological Institute (MNI) space, as implemented in SPM12 (IXI549Space).“

Reviewer #3

We thank the reviewer for his/her positive and helpful comments. We are happy that he/she considers our statistical approaches valid and appropriate and our findings novel and of immediate interest to the field of memory.

Major:

The authors draw several conclusions about emotional modulation of memory, but their behavioral and neuro-imaging findings focus on effects collapsed across valence. For instance, although the results replicate slower forgetting of emotional stimuli, the memory decline from 1d to 28d was weaker in the YOH group than in the PLAC group irrespective of the emotionality of the encoded stimuli. This reviewer thinks that the concluding sentence of the abstract (“potentially explaining the long-term vividness of emotionally arousing memories”) is a bit of a stretch based on the current data that showed effects regardless of stimulus emotionality. Yonelinas and Ritchey (2015) discuss how post-encoding stress does not preferentially slow forgetting of emotional items suggesting post-encoding adrenergic stimulation would influence memory overall, consistent with the present findings.

RESPONSE: We completely agree and based on this comment rephrased the concluding sentence in the abstract. Please see page 2, lines 34 to 36:

„These findings demonstrate that noradrenergic activity shortly after encoding may reverse systems consolidation in humans, thus maintaining vividness of memories over time.”

It appears the fMRI models assessed old and new stimuli based on objective study history, regardless of the participants' memory responses. Are hits and misses analyzed together? False alarms and correct rejections for new stimuli? If so, the authors might consider an explanation and highlight a limitation that some trials included in the analyses might reflect implicit memory processes or false alarms. Memory performance is very high in this study, but if the hit rate was 85% on average, that means that ~10 forgotten images and 2-3 false alarms were analyzed on average in the fMRI models. It would be cleaner to model out misses and false alarms separately from hits and misses on the recognition test in the fMRI models, or clarify/justify this approach.

RESPONSE: As indicated by the reviewer, memory performance was overall very high with on average only a few misses and false alarms. In particular, in the 1d group, recognition performance was, as expected, close to perfect. It was therefore not possible to analyze the neural data separately for hits, misses, correct rejections and false alarms. This analytic strategy would have resulted not only in very few events for some of the regressors, which is a problem for the reliability of the results, but also in the complete loss of some of the participants for the neuroimaging analysis (i.e., those without false alarms or misses). Moreover, due to the expected performance differences between the 1d- and 28d-groups, these issues would have affected in particular the 1d-group and would thus have biased any group comparisons. We therefore decided to analyze our imaging data using the stimulus categories and applied a correlational approach to relate it to behavior. We now clarify this approach further on pages 25 to 26, lines 648 to 651:

„Due to the overall very high memory performance resulting in a low number of misses and false alarms in many participants, we modelled our imaging data based on stimulus category and chose a correlative approach to link these data to behavioral memory performance.“

Further, we discuss this approach as a possible limitation and implications for future studies on pages 19 to 20, lines 486 to 491:

„Another possible limitation refers to the modelling of our imaging data based on the item category regardless of the participants' memory responses. This procedure was chosen because of the overall very high memory performance, specifically in our 1d group, resulting in a low number of false alarms and misses. Future studies on the neural basis of time-dependent changes in memory should employ a design that increases the variability in memory performance, for instance by increasing the number of the to-be-encoded stimuli.“

Further regarding the emotion condition: It is known that the time-dependent emotion advantage tends to be limited to recollection vs. familiarity. Did the authors consider using their confidence data to explore the possibility of the three-way interaction on their higher confidence images? (i.e., high confidence d' based on high confidence hits and fas). The three way interaction of d' appears to be trending (drug \times delay \times emotion: $\beta=0.31$, $p=0.127$) and could be influenced by confidence. Why did the authors not leverage the confidence ratings, at least in a supplementary behavioral analysis?

RESPONSE: We agree and included the confidence ratings now in additional analyses. In a first step, we analyzed whether participants' confidence differed depending on stimulus emotionality, delay or drug. We report the results of this analysis on page 8, lines 181 to 189:

„An additional trial-wise generalized LMM on confidence for hits with drug (PLAC vs. YOH), delay (1d vs. 28d), emotion (neutral vs. negative) and their interactions as fixed effects and the random intercept of participants and stimuli revealed a decrease in confidence in the 28d group, compared to the 1d group ($\beta=-1.91$, $p<0.001$, $z=-5.98$). This expected decrease in confidence in recognizing old items was significantly lower for emotionally negative stimuli (emotion \times delay: $\beta=0.70$, $p=0.007$, $z=2.69$), but not influenced by noradrenergic stimulation (drug \times delay: $\beta=0.63$, $p=0.151$, $z=1.44$). No other main or interaction effects reached significance in this analysis (all $p>0.110$).“

Moreover, in an additional analysis we weighted participants' responses by the level of confidence. This analysis indicated, as before, a significant decrease of memory performance in the 28d-group relative to the 1d-group (main effect delay: $\beta=-1.25$, $t_{123.33}=-5.71$, $p<0.001$). This time-dependent decrease in memory was again significantly lower for YOH than PLAC (drug \times delay: $\beta=0.62$, $t_{123.33}=1.98$, $p=0.0495$). Note that in none of these analyses the interaction drug \times delay \times emotion reached statistical significance ($p>0.091$), despite of a very high power of LMMs. We have added these findings on page 8, lines 189 to 195:

“Moreover, in an additional analysis we weighted participants' responses by the level of confidence. This analysis indicated, as before, a significant decrease in memory performance in the 28d-group relative to the 1d-group (main effect delay: $\beta=-1.25$, 95%-CI[-1.68,-0.82], $t_{123.33}=-5.71$, $p<0.001$). This time-dependent decrease in memory was again significantly lower in the YOH group than in the PLAC group (drug \times delay: $\beta=0.62$, 95%-CI[0.01,1.22], $t_{123.33}=1.98$, $p=0.0495$). Note that in none of these analyses the interaction drug \times delay \times emotion approached statistical significance (all $p>0.091$).“

This paper is particularly timely, given the emotion and reward modulated memory field has become increasingly interested in post-encoding MTL-neocortical consolidation effects, linking to the literature in the discussion might help improve the impact of the field and connect with more of the recent proposed theoretical frameworks. For instance, Clewett and Murty's (2019) influential neuromodulation theory purports that activation of the LC-NE system during post-encoding periods of consolidation will also amplify the preferential processing of salient event features of emotional

stimuli even further, in part through amygdala modulation. Further, Kark and Kensinger have shown increased post-encoding amygdala-hippocampal-occipital resting state functional connectivity relates to behavioral negative memory bias and the degree of occipital reinstatement at 24 hr retrieval. Murty and colleagues (2017) showed that post-encoding mesolimbic-cortical connectivity predicts high reward memory 24 hours after encoding (Murty et al., 2017). What do the authors propose is happening during these post-encoding periods when NA is administered to disrupt these processes (is there a break between the pass-off of the short term synaptic consolidation mode into to systems consolidation mode, keeping memories in the MTL?).

RESPONSE: We agree that the recent work the reviewer is referring to is relevant within the context of the present study and discuss now the theory of Murty et al.¹¹ and the findings of Kark and Kensinger¹² on pages 17 to 18, lines 429 to 436:

„Our findings further align with a recently proposed neuromodulation theory suggesting that activation of the locus coeruleus-norepinephrine system during post-encoding periods of consolidation amplifies the preferential processing of salient event features of emotional stimuli⁴⁶ and the finding that increased post-encoding amygdala-hippocampal-cortical resting state functional connectivity relates to behavioral negative memory bias and the degree of pattern reinstatement after 1d⁴⁸. At the same time, the present findings emphasize the impact of post-encoding noradrenergic arousal on long-term memory, irrespective of valence or arousal of the encoded stimuli.“

Given that our current paradigm did not include any neuroimaging measurements during (initial) consolidation (i.e. after encoding), we can only speculate about the exact mechanism through which YOH altered the systems consolidation. The mechanism proposed by the reviewer represents in our view an interesting alternative and we address this potential mechanism now on page 18, lines 451 to 457:

„Based on these data, it is tempting to speculate that a noradrenergic arousal-related recruitment of the amygdala during initial consolidation may have resulted in a distinct anchoring of memory traces in the hippocampus leading to an increased connectivity between those brain regions, presumably through epigenetically driven transcriptional changes in memory-related genes which may be actively maintained²². At the same time, the burst in noradrenergic stimulation might have led to a break between the pass-off of the short-term synaptic consolidation mode in the hippocampus into a systems consolidation mode, keeping memories in the hippocampus.“

In their discussion the authors touch on the reduced hippocampal involvement at short delays and state that it may be ‘owing to a restructuring that promotes memory maintenance in the long run’. How might reduced (below 0) hippocampal ERS in the YOH group after a short delay facilitate a later boost in hip ERS at 28 days? How and why does NA stimulation increase IFG-hippo FC after 1 day (above 0) in the YOH group? More discussion on the possible reasons and mechanisms of this surprising reversal, beyond slowing, would be helpful.

RESPONSE: This is certainly an important question (and a key question for future research in the area). However, due to the novelty of the finding of a reversed course of memory consolidation due to post-encoding noradrenergic arousal in humans, we can only speculate about the underlying mechanisms. The findings of the previous rodent study¹³ suggest that the shift in systems consolidation dynamics may likely be supported by epigenetically driven DNA methylation and transcriptional changes of memory-related genes in the hippocampus and neocortex. Specifically, norepinephrine-injection into the BLA shortly after learning was associated with a time-dependent decrease in DNA methylation and increase in transcriptional activation of *Reln* in the hippocampus compared to saline. As this gene has been shown to increase synaptic plasticity by increasing long-

term potentiation¹⁴ and support the development of synapses in the hippocampus¹⁵ and its demethylation and transcriptional activation has previously been associated with memory formation¹⁶, such epigenetic mechanisms are likely underlying the reversal in systems consolidation due to post-encoding noradrenergic stimulation. As for our functional connectivity-analyses, it is to be noted, that these are correlational and therefore do not allow inferences about the directionality of effects and might even reflect processes underlying the prevention of a transfer of memory traces. Please see page 18, lines 443 to 450:

„Specifically, norepinephrine-injection into the BLA shortly after learning was associated with a time-dependent decrease in DNA methylation and increase in transcriptional activation of *Reln* in the hippocampus, compared to saline. As this gene has been shown to increase synaptic plasticity by increasing long-term potentiation⁵⁴ and to support the development of synapses in the hippocampus⁵⁵ and its demethylation and transcriptional activation has previously been associated with memory formation⁵⁶, such epigenetic mechanisms are likely underlying the reversing effect of post-encoding noradrenergic arousal on the course of systems consolidation.“

Minor:

Prior work on stress and memory has shown quadratic relationships between post-encoding stress hormone admin and subsequent memory as well as sex differences (Andreano and Cahill, 2006), did the authors consider quadratic relationships? A brief explanation for focusing on the linear effects of stress would be helpful in the paper. Although perhaps underpowered, a mention of the possibility of sex-differences in the limitations section would be helpful.

RESPONSE: In contrast to cortisol, which can be measured from saliva, there is no direct read-out of noradrenergic arousal (except for blood measurements, which would have been arousing and would thus have interfered with our arousal manipulation), which complicates the analysis of quadratic relationships. Alternatively, different dosages of YOH could have been used, which was however simply not feasible given that our design included already four experimental groups and two complex fMRI measurements per participant. Nevertheless, we now discuss this as a relevant aspect for future studies. Please see page 20, lines 492 to 496:

„Furthermore, as prior work on stress and memory has shown quadratic relationships between post-encoding stress hormone administration and subsequent memory²⁶ and it is generally assumed that arousal exerts quadratic effects on cognitive functions⁵⁹, future studies should include different dosages of YOH to further elucidate noradrenergic arousal effects on changes of memory over time.“

In terms of potential sex differences, as mentioned by the reviewer, our sample was not sufficiently powered to test for differences between men and women. Nevertheless, in response to this reviewer's comment we ran an explorative analysis of sex differences. This analysis did not show evidence for sex differences. We added these explorative results on page 8, lines 196 to 202:

„Finally, although our study did not focus on potential differences between men and women and was not sufficiently powered to detect such effects, in light of findings suggesting sex differences in the impact of arousal or stress mediators on memory²⁶, we exploratively analyzed potential sex differences. Including the factor sex into the above LMM did not reveal a significant main effect of sex ($\beta=0.07$, $p=0.820$) nor any interactions with any other factors (all $p>0.384$), suggesting that the effect of post-encoding noradrenergic arousal on memory performance over time was comparable in men and women.“

Images were matched on visual complexity, what kind of visual complexity metric was quantified?

RESPONSE: We did not use a specific visual complexity metric. Old and new items were roughly matched by the first author and an independent rater based on their subjective experience of scene complexity and amount of details. We have clarified this now on page 27, lines 698 to 700:

„(...) old and new items were furthermore roughly matched by the first author and an independent rater based on their subjective experience of scene complexity and number of details (..)“

References

1. Gelman, A. & Hill, J. *Data analysis using regression and multilevel/hierarchical models*. (Cambridge University Press, 2006).
2. Sekeres, M. J., Moscovitch, M. & Winocur, G. Mechanisms of memory consolidation and transformation. in *Cognitive Neuroscience of Memory Consolidation* (eds. Axmacher, N. & Rasch, B.) 17–44 (Springer International Publishing, 2017).
3. Takashima, A. *et al.* Declarative memory consolidation in humans: A prospective functional magnetic resonance imaging study. *Proc. Natl. Acad. Sci. U.S.A.* **103**, 756–761 (2006).
4. Gais, S. Declarative memory consolidation: Mechanisms acting during human sleep. *Learning & Memory* **11**, 679–685 (2004).
5. Brodt, S. & Gais, S. Memory engrams in the neocortex. *Neuroscientist* 107385842094152 (2020) doi:10.1177/1073858420941528.
6. Brodt, S. *et al.* Fast track to the neocortex: A memory engram in the posterior parietal cortex. *Science* **362**, 1045–1048 (2018).
7. Brodt, S. *et al.* Rapid and independent memory formation in the parietal cortex. *Proc. Natl. Acad. Sci. U.S.A.* **113**, 13251–13256 (2016).
8. Kriegeskorte, N., Goebel, R. & Bandettini, P. Information-based functional brain mapping. *Proc. Natl. Acad. Sci. U.S.A.* **103**, 3863–3868 (2006).
9. Norman, K. A. & O'Reilly, R. C. Modeling hippocampal and neocortical contributions to recognition memory: A complementary-learning-systems approach. *Psychol. Rev.* **110**, 611–646 (2003).
10. Eichenbaum, H. On the Integration of Space, Time, and Memory. *Neuron* **95**, 1007–1018 (2017).
11. Clewett, D. & Murty, V. P. Echoes of emotions past: How neuromodulators determine what we recollect. *eNeuro* **6**, ENEURO.0108-18.2019 (2019).
12. Kark, S. M. & Kensinger, E. A. Post-encoding amygdala-visuosensory coupling is associated with negative memory bias in healthy young adults. *J. Neurosci.* **39**, 3130–3143 (2019).
13. Atucha, E. *et al.* Noradrenergic activation of the basolateral amygdala maintains hippocampus-dependent accuracy of remote memory. *Proc. Natl. Acad. Sci. U.S.A.* **114**, 9176–9181 (2017).
14. Beffert, U. *et al.* Modulation of synaptic plasticity and memory by Reelin involves differential splicing of the lipoprotein receptor Apoer2. *Neuron* **47**, 567–579 (2005).
15. Niu, S., Yabut, O. & D’Arcangelo, G. The Reelin signaling pathway promotes dendritic spine development in hippocampal neurons. *J. Neurosci.* **28**, 10339–10348 (2008).
16. Miller, C. A. & Sweatt, J. D. Covalent modification of DNA regulates memory formation. *Neuron* **53**, 857–869 (2007).

REVIEWERS' COMMENTS

Reviewer #1 (Remarks to the Author):

The authors have addressed all my previous questions satisfactorily. I do not have any further issues.

Reviewer #2 (Remarks to the Author):

I would like to thank the authors for the effort they have put into addressing my questions. I have no further comments. I am fully in line with using the revised title, thank you for additionally clarifying in the main text what "reversed" is referring to. I also appreciate that additional analyses have been performed and are now reported in the main text and supplement. I agree with the authors to report the 3mm searchlight results since the larger radius looks consistent, and I follow their reasoning on hippocampal volume, with the hipp being a major ROI in this study.

I have nothing more to add except: Congrats to the authors on their highly interesting findings!

Reviewer #3 (Remarks to the Author):

The authors did a fantastic job addressing each of my concerns, with added discussion, clarification, and analyses. Along with the revisions addressing the comments of R1 and R2, the manuscript has improved substantially and will be of great interest to the field.